



# Observation operators for assimilation of satellite observations in fluvial inundation forecasting

Elizabeth S. Cooper[1], Sarah L. Dance[1,2], Javier García-Pintado[3], Nancy K. Nichols[1,2], and Polly Smith[1]

[1]Department of Meteorology, University of Reading,UK.
[2]Department of Mathematics and Statistics, University of Reading, UK.
[3]MARUM Center for Marine environmental Sciences and Department of Geosciences, University of Bremen, Germany.

**Correspondence:** Elizabeth Cooper (e.s.cooper@pgr.reading.ac.uk)

**Abstract.** Images from satellite-based synthetic aperture radar (SAR) instruments contain large amounts of information about the position of flood water during a river flood event. This observational information typically covers a large spatial area, but is only relevant for a short time if water levels are changing rapidly. Data assimilation allows us to combine valuable SAR-derived observed information with continuous predictions from a computational hydrodynamic model and thus to produce a

better forecast than using the model alone. In order to use observations in this way a suitable observation operator is required. In this paper we show that different types of observation operator can produce very different corrections to predicted water levels; this impacts on the quality of the forecast produced. We discuss the physical mechanisms by which different observation operators update modelled water levels and introduce a novel observation operator for inundation forecasting. The performance of the new operator is compared in synthetic experiments with that of two more conventional approaches. The conventional

approaches both use observations of water levels derived from SAR to correct model predictions. Our new operator is instead designed to use backscatter values from SAR instruments as observations; such an approach has not been used before in an ensemble Kalman filtering framework. Direct use of backscatter observations opens up the possibility of using more information from each SAR image and could potentially speed up the time taken to produce observations needed to update model predictions. We compare the strengths and weaknesses of the three different approaches with reference to the physical mechanisms

by which each of the observation operators allow data assimilation to update water levels in synthetic twin experiments in an idealised domain.

## 1 Introduction

During a fluvial flood it is possible to use a numerical hydrodynamic model to predict future water levels and flood extents. Such predictions are subject to uncertainties and can be inaccurate; data assimilation can therefore be used to improve predictions by

updating model forecasts based on various types of observations (e.g. Lai and Monnier (2009), Matgen et al. (2007), Garcia-Pintado et al. (2013), Garcia-Pintado et al. (2015), Ricci et al. (2011), Barthélémy et al. (2016) and Schumann et al. (2009)). For flooding, useful observations of river flow rate or water depth could come from river gauges. However the number of gauges is declining worldwide (Vörösmarty et al. (2001)) and a method that can work in ungauged catchments is therefore desirable.





For this reason satellite images, and in particular synthetic aperture radar (SAR) images of flooded areas can be a good source of information (Grimaldi et al. (2016)).

SAR sensors are active, side-looking sensors included on several satellites, e.g. CosmoSkymed and Sentinel 1. Radiation (of wavelength $cm$ to $m$) is emitted from the satellite and directed towards the surface of the Earth. The returning signal is

recorded at a sensor and can be used to reconstruct information about the observed terrain. SAR radiation is cloud penetrating, giving the instruments all-weather capability. SAR instruments can also produce observations day and night, unlike passive sensors that rely on solar radiation.

The strength of the returned signal measured at the SAR sensor depends strongly on the roughness properties of the surface from which it has been reflected. During a flood event SAR images therefore generally show a clear difference between flooded

and non-flooded areas. Pixels in flooded or other wet areas such as lakes and rivers have low backscatter values and appear as dark areas on SAR images; dry areas have higher backscatter values and dry pixels therefore appear paler. There are a number of techniques for separating pixels into wet and dry areas based on backscatter. Methods include thresholding (e.g. Henry et al. (2006)) with varying levels of user-interpretation (as compared in Brown et al. (2016)), region growing/clustering ('snakes') (e.g. Horritt et al. (2001)) and change detection (e.g. Hostache et al. (2012)). These techniques can be used to

provide observational information for data assimilation frameworks, but are also used for flood mapping and monitoring (as in e.g. Brown et al. (2016), Matgen et al. (2011)) and for validation and calibration of inundation models (e.g. Mason et al. (2009b), Baldassarre et al. (2009), Wood et al. (2016)). In the case of model calibration, Mason et al. (2009a) and Stephens et al. (2013) suggest that comparing modelled and observed derived water level measures from SAR images results in better calibration than when using binary wet-dry comparisons. However, it is not clear whether derived water levels provide better

observation impact than wet/dry observations in data assimilation.

In this work we consider different ways in which information from a SAR image can be used to correct inundation forecasts using data assimilation. The use of observations requires two steps. First, we must extract relevant, useable information from a SAR image. This involves processing the raw SAR data in some way to produce an observation, or set of observations, per image. In the second step we need to use an observation operator to map our model state vector into observation space - i.e. we

extract the equivalent information from our model in order to compare it to the observations. The size of the difference between the observation and the equivalent information from the model forecast is then used to calculate an update or correction to the forecast. The observation operator depends on the type of observational information used and we show in this paper that the impact of observations on the forecast can be strongly dependent on the observation operator approach used. Despite this, the mechanisms through which different observation types and different observation operators update hydrodynamic forecasts

have not received much attention in the literature.

In order to extract observational information from a SAR image, authors such as Mason et al. (2012), Giustarini et al. (2011), Neal et al. (2009) and Matgen et al. (2007) have used an approach which relies on identifying the flood edge. Terrain information, e.g. from a digital terrain model, is then used to infer information about water levels on the floodplain. Water level observations (WLOs) can then be compared with model forecast water levels. Examples of two observation operators using

flood edge WLOs are described further in section 3. A different type of observation is used for data assimilation in Wood (2016)





and Hostache et al. (2018), in which flood probability maps are produced from SAR images using the method in Giustarini et al. (2016). Particle filter data assimilation techniques are then used to update a hydrodynamic model using flood probability maps as observations.

We propose a new type of observation operator which directly uses pixel-by-pixel backscatter values as observations. As in Wood (2016) and Giustarini et al. (2016), we rely on the fact that SAR images yield distinct distributions of wet and dry backscatter values. However, our method employs an ensemble transform Kalman filter (ETKF) approach with a novel observation operator; we directly use measured SAR backscatter values as observations rather than derived flood probability measures.

In this paper we examine the performance of our new observation operator and that of two flood-edge observation operators in a series of synthetic experiments. We compare the physical mechanisms by which the different approaches update predicted water levels in the ETKF; to the authors' knowledge these physical mechanisms have not been discussed in the literature before. We outline the ETKF data assimilation algorithm in section 2 and in section 3 we describe the three observation operators which we have compared. Further details of our experiments are given in section 4, including an outline of the hydrodynamic model. In section 5 we demonstrate how well the assimilation can update model forecast water levels towards the truth with each observation operator and explore the different physical mechanisms by which updates are made. We also test the ability of the three operators to successfully update the model channel friction parameter through an augmented state vector approach. We find that our new backscatter operator generates better corrections to the state and parameter values than the simplest approach to assimilating flood edge observations, but does not always perform as well as the 'nearest wet pixel' approach. In section 6 we conclude with a comparison of the relative strengths and weaknesses of the three different observation operators.

## 2 Data assimilation

In this paper we explore the use of observations from SAR images in updating forecasts from a hydrodynamic flood model. In section 2.1 we outline the ETKF data assimilation framework we use in our experiments (Bishop et al. (2001)). In section 2.2 we describe the joint state-parameter estimation method we use to update the channel friction parameter value.

### 2.1 Ensemble transform Kalman filter (ETKF)

In data assimilation, forecasts from a numerical model are combined with observations of the same system. We use a state vector, $\mathbf{x}(t_k) \in \mathbb{R}^N$ to represent the state of the dynamical system at time $t_k$. Here, our model domain is split into $N$ computational cells and the state vector contains $N$ water depths at a given time. In this paper we use an ensemble of state vectors, where each state vector in the ensemble represents a possible state of the system. For an ensemble made up of $M$ state vectors (members), $\mathbf{x}_i$, $(i = 1, 2, ..., M)$ the best estimate of the true state of the system is represented by the mean state, $\bar{\mathbf{x}}$, where

$$\bar{\mathbf{x}} = \frac{1}{M} \sum_{i=1}^{M} \mathbf{x}_i. \tag{1}$$




We can define a perturbation matrix, $\mathbf{X} \in \mathbb{R}^{N \times M}$, that can be used to derive a measure of uncertainty in the mean state. The perturbation matrix is

$$\mathbf{X} = \frac{1}{\sqrt{M-1}}(\mathbf{x}_1 - \bar{\mathbf{x}} \;\; \mathbf{x}_2 - \bar{\mathbf{x}} \;\; ...... \;\; \mathbf{x}_M - \bar{\mathbf{x}}). \tag{2}$$

We can then express the ensemble error covariance matrix, $\mathbf{P} \in \mathbb{R}^{N \times N}$ as

$$\mathbf{P} = \mathbf{X}(\mathbf{X})^T. \tag{3}$$

The ETKF is a two-step sequential data assimilation technique. In the forecast step, each vector $\mathbf{x}_i$, is evolved in time using the forecast equation,

$$\mathbf{x}_i(t_{k+1}) = \mathfrak{M}(\mathbf{x}_i(t_k)), \tag{4}$$

where $\mathfrak{M}$ is the forecast model. Here, $\mathfrak{M}$ is a hydrodynamic model built using Clawpack code (see section 4.1); the model

evolves water levels in each ensemble member with time, generating an ensemble of flood forecast realisations.

In the update step the mean state vector and the error covariance matrix are both updated based on observational information. We assume that the observations are related to the true state of the system, $\mathbf{x}^t$ according to

$$\mathbf{y}_{obs} = \mathbf{h}(\mathbf{x}^t) + \epsilon, \tag{5}$$

where the vector $\mathbf{y}_{obs} \in \mathbb{R}^p$ contains $p$ observations. The vector $\epsilon$ represents observation error, which we assumed to be unbiased

and stochastic with covariance $\mathbf{R} \in \mathbb{R}^{p \times p}$. The observation operator, $\mathbf{h} : \mathbb{R}^N \to \mathbb{R}^p$ maps the state vector into observation space. If the state vector and the observation vector contain the same quantity (e.g. water depth) then the observation operator is generally just required to pick out the values in the state vector corresponding to the spatial position of the observation(s); this may involve spatial interpolation if observations are not located at model grid points. However, it is commonly the case that observations are different quantities to those in the state vector and the observation operator therefore contains information

about how the observed and state vector quantities are related as well as positional information. Different observation types (e.g. water elevation or wet/dry pixel information) therefore require different observation operators for the same model (i.e. for the same state vector).

In order to update the model forecast it is useful to create a forecast-observation ensemble, which contains $M$ forecast-observation vectors, $\mathbf{y}_i^f$, $(i = 1, 2...M)$ such that

$$\mathbf{y}_i^f = \mathbf{h}(\mathbf{x}_i^f). \tag{6}$$

Equation (6) shows that the observation operator, $\mathbf{h}$, is applied to each state vector in order to extract observation equivalent information; each forecast-observation vector, $\mathbf{y}_i^f \in \mathbb{R}^p$ is what would be observed if the corresponding state vector, $\mathbf{x}_i^f$





represented the true state of the system. The model equivalent of the observation vector is given by the mean of the forecast-observation ensemble, $\overline{\mathbf{y}^f} \in \mathbb{R}^p$, where

$$\overline{\mathbf{y}^f} = \overline{\mathbf{h}(\mathbf{x})} = \frac{1}{M} \sum_{i=1}^{M} \mathbf{h}(\mathbf{x}_i). \tag{7}$$

Note that when the observation operator is nonlinear,

$\overline{\mathbf{h}(\mathbf{x})} \neq \mathbf{h}(\bar{\mathbf{x}}). \tag{8}$

We can also define a perturbation matrix $\mathbf{Y}^f \in \mathbb{R}^{p \times p}$ for the forecast-observation ensemble matrix:

$$\mathbf{Y} = \frac{1}{\sqrt{M-1}} (\mathbf{y}_1 - \bar{\mathbf{y}} \ \ \mathbf{y}_2 - \bar{\mathbf{x}} \ \ ...... \ \ \mathbf{y}_M - \bar{\mathbf{y}}). \tag{9}$$

The mean state vector and error perturbation matrix are updated separately in the ETKF. The mean state is updated according to

$\overline{\mathbf{x}^a} = \overline{\mathbf{x}^f} + \mathbf{K}(\mathbf{y}_{obs} - \overline{\mathbf{y}^f}), \tag{10}$

where $\overline{\mathbf{x}^a} \in \mathbb{R}^N$ and $\overline{\mathbf{x}^f} \in \mathbb{R}^N$ are the means of the analysis and forecast ensemble respectively. The ETKF uses an ensemble version of the Kalman gain, $\mathbf{K} \in \mathbb{R}^{N \times p}$ is, as defined in equation (13). The ensemble Kalman update (10) can be written in terms of the innovation, $\boldsymbol{\delta}_y$, where

$$\boldsymbol{\delta}_y = \mathbf{y}_{obs} - \overline{\mathbf{y}^f}. \tag{11}$$

The innovation is calculated in observation space. The term

$$\mathbf{K}(\boldsymbol{\delta}_y) \tag{12}$$

is known as the increment, and is the difference between $\overline{\mathbf{x}^a}$ and $\overline{\mathbf{x}^f}$. The increment is calculated in state space.

We use a square root formulation for the ETKF, following Livings et al. (2008) and Livings (2005). This formulation is also used in Garcia-Pintado et al. (2013) and Cooper et al. (2018). In this approach the ensemble version of $\mathbf{K}$ is written as

$\mathbf{K} = \mathbf{X}^f (\mathbf{Y}^f)^T (\mathbf{Y}^f (\mathbf{Y}^f)^T + \mathbf{R})^{-1}. \tag{13}$

The state error perturbation matrix is updated in the ETKF according to

$$\mathbf{X}^a = \mathbf{X}^f \mathbf{T}. \tag{14}$$

The perturbation matrix is updated by the matrix $\mathbf{T} \in \mathbb{R}^{M \times M}$. The matrix $\mathbf{T}$ is constructed in a way that ensures that the analysis state error covariance, $\mathbf{P}^a = \mathbf{X}^a (\mathbf{X}^a)^T$ is the same as the anaylis error covariance calculated in the Kalman covariance

update (in e.g. Kalman (1960)). See Cooper et al. (2018) for more details of how $\mathbf{T}$ is computed.





## 2.2  Joint state-parameter estimation

State augmentation techniques can be used to correct values of uncertain forecast model parameters at the same time as the state is updated. In this approach, parameters are appended to the state vector ( see Smith et al. (2013); Navon (1998); Evensen et al. (1998); Smith et al. (2009, 2011)), producing an augmented state vector, $\mathbf{x}_{aug}$:

$$\mathbf{x}_{aug} = \begin{bmatrix} \mathbf{x} \\ \mathbf{b} \end{bmatrix}, \tag{15}$$

where $\mathbf{x}_{aug} \in \mathbb{R}^{N+q}$. The vector $\mathbf{b} \in \mathbb{R}^{q}$ contains $q$ parameters. In this paper only one parameter is being updated, so that $\mathbf{b}$ is scalar. The parameter we are updating in this paper is the Manning's friction coefficient in the river channel, $n_{ch}$, as the evolution of a flood is known to be very sensitive to this parameter.

The forecast equation for the case of an augmented state vector can be written as

$$\mathbf{x}_{aug}(t_{k+1}) = \begin{bmatrix} \mathfrak{M}(\mathbf{x}(t_k)) \\ \mathbf{b}(t_k) \end{bmatrix}. \tag{16}$$

Equation 16 shows that we assume the value of $n_{ch}$ remains constant during the forecast step and changes only when the update equation is applied.

The augmented state vector is updated by the ETKF algorithm through equations (10) and (14). Parameter value(s) are updated according to the observations due to covariances between errors in the model state and errors in the parameter(s).

Model friction parameter values are more traditionally calculated using offline calibration techniques and data from previous flood events. Updating parameter values using a state augmentation approach has the advantage that it uses information from observations of the flood event of interest as it occurs. State augmentation can therefore take into account any recent changes to the river and its environment.

## 3  Observation operators for inundation forecasting

Much existing work on data assimilation for fluvial inundation forecasting has focussed on assimilating derived water level observations. Water level extraction is based on the fact that it is usually possible to differentiate between wet and dry areas in a SAR image; the contrast in backscatter between wet and dry pixels means that it is therefore possible to determine the position of the edge of a flooded area. Along this edge, the water elevation is the same as the elevation of the topography. This means that as long as a flood edge can be accurately identified and topographical information is available (e.g. a digital terrain model
(DTM)), water levels at the flood edge can be derived from a SAR image. This approach has also been used for operational flood mapping, e.g. Brown et al. (2016). In practise, it is not possible to accurately determine flood extents from SAR images over the whole 'edge' of a flooded area. This is clearly shown in Mason et al. (2012) and can lead to few, sparse observations of this type.





In the remainder of this section we describe the three different observation operators used in this study. In section 3.1 we describe the simplest way to use flood edge water level observations; the results in section 5.1.2 illustrate the problems with this approach. Section 3.2 gives an outline of the more sophisticated approach to using water level observations used in Garcia-Pintado et al. (2013) and Garcia-Pintado et al. (2015). In section 3.3 we describe our new observation operator.

## 3.1 Observation operator $\mathbf{h}_s$: simple flood edge assimilation

In this approach, we assume $\mathbf{y}_{obs}$ comprises $p$ water level observations at flood edge positions. The simplest way to use these observations to calculate an innovation is to extract water level information from each ensemble member at each observed flood edge location. The observation operator in this approach, $\mathbf{h}_s$, picks out water level predictions at the positions of the observed flood edges for each ensemble member. Some method of interpolation will generally be necessary in order to locate the closest

cell to the measured flood edge location, but this was not needed in our identical twin experiments as the truth and forecast simulations use the same grid. The simple observation operator $\mathbf{h}_s$ in our case is therefore described by a sparse matrix, $\mathbf{H}_s$ dimension ($p$ by $N$) containing one and zero values such that water elevation predictions corresponding to the positions of flood edge observations are mapped with weight one and all other values with weight equal to zero, i.e.

$$\mathbf{h}_s(\mathbf{x}_i^f) = \mathbf{H}_s\mathbf{x}_i^f. \tag{17}$$

The value of $\overline{\mathbf{y}^f}$ is then calculated according to equation (7).

This approach can lead to problems in application and is therefore not widely used, but we include it here to show the importance of how observations are used in data assimilation. The problem with this simple method is essentially that it does not use all of the available information. All ensemble members that predict shallower local water levels than the truth at the position of the observation will make the same contribution to $\overline{\mathbf{y}^f}$; they will all predict zero water depth at the flood edge

position no matter how much shallower the ensemble prediction is than the truth.

## 3.2 Observation operator $\mathbf{h}_{np}$: nearest wet pixel approach

In this approach we assume again that $\mathbf{y}_{obs}$ comprises $p$ water level observations at flood edge positions. In Garcia-Pintado et al. (2013) and Garcia-Pintado et al. (2015) the authors use flood edge water level observations with a more sophisticated observation operator, referred to here as the 'nearest wet pixel' method. The new observation operator, $\mathbf{h}_{np} \in \mathbb{R}^{p \times N}$ can be

described as a sparse matrix containing values of one and zero, so that

$$\mathbf{h}_{np}(\mathbf{x}_i^f) = \mathbf{H}_{np}\mathbf{x}_i^f. \tag{18}$$

Now however, water elevation values are mapped differently. Each row of $\mathbf{h}_{np}$ contains a one at the positions corresponding to a flood edge locations observation only if the corresponding water elevation value in $\mathbf{x}_i^f$ is greater than the observed flood edge elevation. Where this is not the case, the entry in $\mathbf{x}_i^f$ corresponding to the 'nearest wet pixel' (i.e. the local flood edge position




as predicted by the $i$th ensemble member) is instead given a weighting of one. Unlike the simple approach, this method allows information to be included from ensemble members that predict shallower water levels than the truth, since the contribution to $\overline{\mathbf{y}^f}$ will depend on the position of the flood edge predicted by each shallower ensemble member.

Finding the 'nearest wet pixel' can be difficult in practice, since is it important to find the local flood edge that corresponds
to the observation. In simplified topography such as used in this study, this can be assumed to be the first wet pixel encountered when moving from the observation towards the centre of the river along a cross section perpendicular to the flow of the river. In situations where the topography is complex (e.g. the local direction of flow is not clear, or the river has tight meanders) finding the nearest wet pixel becomes more complicated. One approach is to require that the nearest wet pixel is in the direction of the steepest downhill descent from the observation location.

A related approach has been successfully used by Matgen et al. (2007), Giustarini et al. (2011), Neal et al. (2009) and Matgen et al. (2010), in which it is assumed that the water level measured at a flood edge can be used to define the water level along the whole horizontal cross section of river valley perpendicular to the flow of the river. In other words, the observed water elevation at the flood edge is extrapolated across the river valley in a direction perpendicular to the flow of the river. Again, this could potentially cause problems in situations in which the local direction of flow is not clear or the river has tight meanders.
There may also be problems if the observations relate to bodies of water on the floodplain that have become hydraulically separate from the river when the flood is receding; such ponding was observed in the floods of the Severn and Avon rivers near Tewkesbury, UK in 2014 (Waller et al. (2018)).

### 3.3  New observation operator, $\mathbf{h}_b$: backscatter approach

We have developed an alternative method for extracting observations from a SAR image, which directly uses SAR backscatter
measurements as observations, rather than derived water elevation information. This means that the observation vector $\mathbf{y}_{obs}$ comprises $p_b$ backscatter values at a number of selected pixel locations. The method potentially allows for more information to be used per SAR image, as information can be used from areas excluded from water elevation calculations. This could reduce the time taken to process a SAR image and produce useable observations.

The observations used in this method are measured SAR backscatter values; we follow the approach of Giustarini et al.
(2016) in assuming that the backscatter values from a SAR image can be characterised as belonging to two separate probability density functions; one for wet pixels and one for dry pixels. We assume that we can create a histogram of backscatter values in the area of interest (Giustarini et al. (2016)). Two Gaussian curves are then fitted to the histogram, corresponding to the wet and dry probability density functions. The distribution of wet pixels has a mean backscatter value $m_w$ and variance $\sigma_w^2$. The distribution of dry pixels has mean and variance $m_d$ and $\sigma_d^2$. Dividing the SAR image into tiles may be necessary for this to
work optimally; otherwise the distribution of dry pixels is likely to dominate the histogram and make the wet pixel distribution difficult to resolve (see e.g. Chini et al. (2017)).

A new observation operator is required in order to use backscatter observations in data assimilation. The operator needs to take each state vector (containing water levels in each pixel) and transform that information into model equivalent backscatter values. This could potentially be achieved using a SAR simulator to generate a synthetic SAR image, but this would be





computationally expensive and would require detailed knowledge of the underlying terrain and land-use cover. Instead we take a statistical approach that makes use of the wet and dry pixel backscatter distributions obtained from a SAR image. The observation operator comprises two steps. We can describe this such that

$$\mathbf{y}_i^f = \mathbf{h}_b(\mathbf{x}_i^f) = \mathbf{h}_{b2}(\mathbf{H}_{b1}\mathbf{x}_i^f), \tag{19}$$

where $\mathbf{H}_{b1}$ is a sparse matrix, dimension $(p_b \times N)$ which extracts values corresponding to observation location positions; each row contains a 1 at positions corresponding to backscatter observation locations and all other values are zero. The non linear operator $\mathbf{h}_{b2}$ is then applied to $\mathbf{H}_{b1}\mathbf{x}_i^f \in \mathbb{R}^{p_b}$. This operation transforms each entry in the vector $\mathbf{H}_{b1}\mathbf{x}_i^f$ into $m_w$ if water is predicted in the cell, or $m_d$ if the cell is predicted to be dry. As for the other observation operators, interpolation will be necessary when observed backscatter cells do not correspond to the positions of model forecast information. As already mentioned, this was not necessary in our synthetic study as we used the same model to generate both the forecast values and synthetic observations; cell locations were therefore the same. The observation equivalent forecast vector is then given by

$$\overline{\mathbf{y}^f} = \frac{1}{M}\sum_{i=1}^{M}\mathbf{h}_{b2}(\mathbf{H}_{b1}\mathbf{x}_i^f). \tag{20}$$

This method potentially allows the use of more observations: in general the number of available backscatter values from a SAR image, $p_b$ is much larger than the number of reliable flood edge observations.

## 4 Experimental design

### 4.1 Hydrodynamic model

The inundation model used in this work is a non-linear hydrodynamic model. The model uses Clawpack code (Clawpack Development Team (2014), Mandli et al. (2016), LeVeque (2002)) to solve the two dimensional shallow water equations in order to simulate water flowing in a channel and overtopping onto a flood plain. Clawpack solves the shallow water equations using Riemann solvers and finite volume methods, and is able to simulate the wet-dry interfaces that occur during a flood George (2008)). The software considers the domain of interest as a user-defined number of cells, $N$, and calculates changes in depth and velocity of the water in each cell. Clawpack uses a source term in the momentum equation to model friction effects. Momentum reduction depends on a user-specified Manning's friction coefficient. Our experiments required an inflow source term to model water arriving in the river from upstream; we added this functionality to the Clawpack code, see Cooper et al. (2018) for details. The time step for the calculations is automatically adjusted to preserve numerical stability.

### 4.2 Domain

Experiments to compare the performance of the three operators have been carried out in an idealised river valley-like domain. The use of an idealised domain is important here so that we can examine the effects of the operators under ideal conditions,

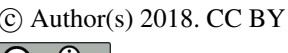



without the complications of complex topography. It will also be important to understand how the operators work under real conditions, but experiments in an idealised topography are a vital first step.

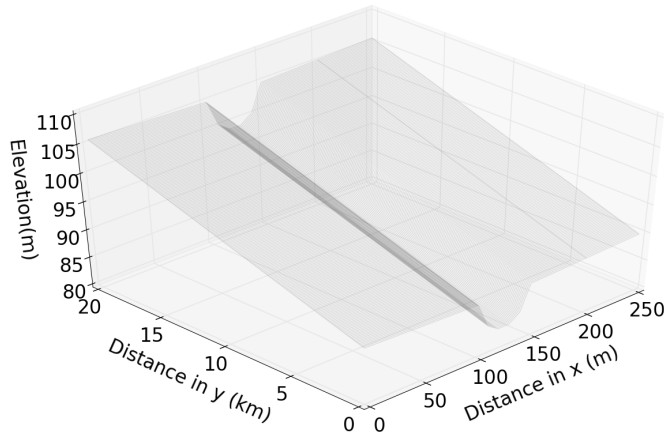

**Figure 1.** Test domain used in all assimilation experiments.

The test domain used in the experiments in this paper is the same as that used in Cooper et al. (2018) and is shown in figure 1. The domain has dimensions of 20km by 250m and describes a gently sloping valley and river channel (with upstream-downstream slope of 0.08%). The domain is split into grid cells of size 10m by 10m for computation. The river channel is prescribed to be the central 5 grid cells in the $x$ direction for all values of $y$ and is 50m wide; the flood plain is defined as the rest of the domain. The slope of the floodplain towards the river is 0.8% based on values derived from a DTM of a stretch of the river Severn in the U.K.

## 4.3 Twin experiments

We have carried out a number of twin experiments in order to illustrate and compare how well forecasts can be corrected when using the three different observation operator approaches. The experiments use a 'truth' flood simulation and a forecast ensemble of flood realisations comprising 100 members. The forecast ensemble is updated using synthetic observations at several times during the simulation time; synthetic observations are created from the truth as described in section 4.4. The analysis water levels (and parameter values) can then be compared to the true water levels (and parameter values) to see how well the assimilation corrects the forecast.

In this work, the truth flood is driven by a time-varying inflow based on data taken from a gauge on the River Severn during a flood in November-December 2012. The true inflow is shown in figure 2; the figure also shows the inflows driving the ensemble members. All the inflows used here were also used in the experiments reported in Cooper et al. (2018) Inflows for each ensemble member were generated by perturbing the true inflow with additive, time correlated random errors. Time

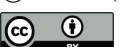


correlated errors were generated for each ensemble inflow using a first order autoregression (AR(1)) technique (Wilks (2011)) with zero mean, according to

$$e_{i,0} = w_{i,0},$$

$$e_{i,k} = r e_{i,k-1} + (1-r^2) w_{i,k}, \tag{21}$$

where $e_{i,k}$ is the error added to the inflow at the $k$th timestep in the $i$th ensemble member. The term $w_{i,k}$ is taken from a normal distribution $\mathcal{N}(0, 0.15 \times \text{true inflow})$; $i$ refers to ensemble member and $k$ refers to the timestep. The autocorrelation coefficient, $r < 1$ was set to 0.997; this very high coefficient means that the errors are close to persistent in time for each ensemble member and that each inflow ensemble member is smooth. The standard deviation of the random part of the error corresponds to the value used to generate inflow errors in Garcia-Pintado et al. (2015) and results in inflows that fit within the range given in Di Baldassarre and Montanari (2009) (4% to 43%). The mean of the inflow ensemble has negligible bias relative to the true inflow. The experiments shown here all use the same inflow for the truth and the same set of perturbed inflows for the forecast ensemble. For a different true inflow and different ensemble inflow error realisations, the results obtained using the different observation operators may compare slightly differently. However, the mechanisms we describe would be the same.

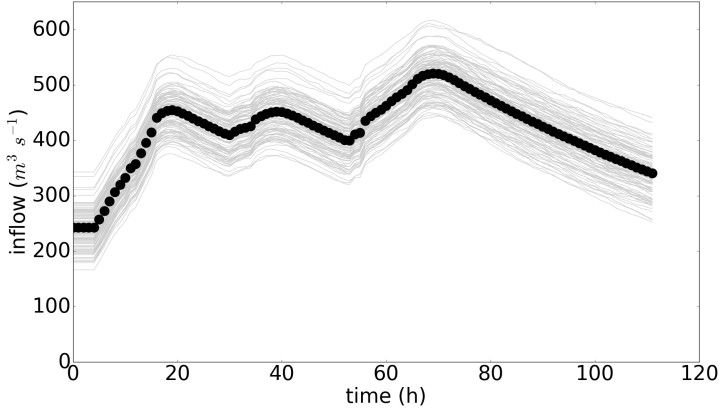

**Figure 2.** Inflows with time. True inflow values are represented with circles and ensemble inflows are shown by grey lines.

Each ensemble member was run with a different value of the channel friction parameter, $n_{ch}$. The behaviour of flood water is highly sensitive to $n_{ch}$ (Hostache et al. (2010), James et al. (2016)), with low channel friction parameter values leading to water travelling through and leaving the domain more quickly. This leads to shallower water levels (and less flooding) in our simple domain for a given inflow. Conversely, higher channel friction parameter values lead to water moving slowly through the domain, leading to deeper water levels in the channel and more flooding. We chose a true value of $n_{ch} = 0.04$, equal to the value for a natural stream given in Maidment and Mays (1988). The value of $n_c h$ for each forecast ensemble member was initially drawn from a normal distribution with a mean that is different to the true value. This imposed bias in the forecast ensemble channel friction parameter means that we can test how well data assimilation with different observation operators can



correct the forecast state and parameter value towards the truth. Using an incorrect channel friction parameter in the forecast is realistic, as the true value is unlikely to be known in operational situations. Forecast channel friction parameters are randomly drawn from $\mathcal{N}(0.05, 0.01)$ for experiments with positive bias in $n_{ch}$ and $\mathcal{N}(0.03, 0.01)$ for experiments with negative bias in $n_{ch}$. The true value of $n_{ch}$ falls within one standard deviation of the mean of each initial $n_{ch}$ distribution and our choices

of friction parameter values fit with the range used in Horritt and Bates (2002). On the flood plain the value of the friction parameter is likely to be higher than $n_{ch}$ due to the effects of vegetation. In this paper we used a true value for the flood plain friction parameter of $n_{fp} = 0.05$; the same, true value for $n_{fp}$ was used for each ensemble member. The value of this parameter is likely to have an impact on the dynamics of a flood event, but flooding is commonly understood to be less sensitive to $n_{fp}$ than $n_{ch}$ (e.g. Hostache et al. (2010)). Here we focus on the ability of the observation operators to update $n_{ch}$ only.

## 4.4 Synthetic observations

In identical twin experiments, observations are generated from a truth run; in this case the 'truth' flood simulation is described in section 4.3. For the two conventional observation operators we selected six synthetic observations of water elevation at the true flood edge at $y = 500m, 700m, 900m, 1100m, 1300m, 1500m$. The water elevation at these points is directly available from the state vector of water levels provided by our truth run. Each synthetic observation mimics a SAR-derived water level

observation at a given cross section by locating a flood edge and using the true, calculated water elevation at this position as the observation. Here we define the flood edge WLO to be the elevation at the first 'dry' pixel encountered when moving in a perpendicular direction from the centre of the channel along one of our defined cross sections. (We use observations on the left hand side of the domain, i.e. where $x < 125m$, but since the domain and inflows are symmetrical in our simple experiments this choice is arbitrary; we could have instead used observations from the right hand side of the channel, or a combination

of the two.) We added unbiased, Gaussian noise with a standard deviation of 0.25m to each observation; this is the same as the observation error used by Garcia-Pintado et al. (2015) in a case study. Observation error may be due to SAR instrument error or errors in determination of flood extent. The spacing of 200m between observations represents an optimistic best case situation, and is the same as the smallest recommended distance between thinned flood edge values for use in an assimilation system in Mason et al. (2012) (note that the other selection criteria used in the paper are not applicable here due to the use of

synthetic observations). In fact, more recent work suggests a much longer correlation length scale between observation errors in a real case study (Waller et al. (2018)), in which the authors point out that part of the observation error correlation is due to the observation operator.

 In order to test our backscatter observation operator we require synthetic backscatter observations; we therefore create a synthetic SAR image from our truth run, comprising backscatter values in each cell. We can then extract synthetic backscatter

observations at desired locations. We have taken a very simple approach to generating a simplified synthetic SAR image in order to perform proof-of-concept experiments with our new observation operator; we will apply the method to a real case study and real SAR images at a later date. To generate a synthetic SAR image, we have taken our truth run water level output and applied a threshold water level of 5cm in each cell to determine which cells are wet and which are dry. Water levels below a threshold of a few cm are likely to be misclassified as dry in a real SAR image due to vegetation. Synthetic backscatter values





are then assigned to each cell: dry cells are assigned a backscatter value drawn from $\mathcal{N}(m_d, \sigma_d^2)$ and wet cells a value from $\mathcal{N}(m_w, \sigma_w^2)$. For this, we have used values of $m_w = -14.84$, $\sigma_w = 2.25$, $m_d = -8.59$ and $\sigma_d = 1.53$, which are experimentally derived from a SAR image in Giustarini et al. (2016). An example simplified synthetic SAR image, generated from the truth run at $t = 40h$, is shown in figure 3.

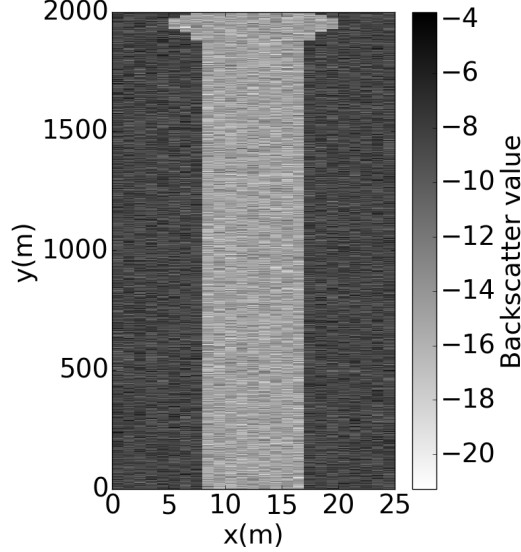

**Figure 3.** Synthetic SAR image generated from truth run water levels as described in section 4.4

In order to derive synthetic observations from the synthetic SAR image, the observation process is then carried out, i.e. we

– bin all the synthetic backscatter values in a histogram - see figure 4

– fit two Gaussian curves to the synthetic backscatter values (using python fitting algorithm scipy.optimize.curve_fit) - see figure 4

– extract new values of $m_{w1}, \sigma_{w1}, m_{d1}$ and $\sigma_{d1}$ from these distributions; these values are naturally very similar to the
experimental values used to create the synthetic SAR image. We use a different realisation of observation error for each synthetic image (i.e. at each observation time); typical values of $m_{w1}, \sigma_{w1}, m_{d1}$ and $\sigma_{d1}$ are within 1% of $m_w, \sigma_w, m_d$ and $\sigma_d$.

We then extract backscatter values to be synthetic observations. Although it would be possible to use a large number of backscatter observations in this method, for the experiments presented here we have not used all of the available synthetic
observations. There are a number of reasons for limiting the number of observations. Firstly, observation errors are likely to be correlated for observations that come from positions close to each other in physical space. Some thinning of the observations is therefore necessary to meet the requirement that the observations used in the assimilation have uncorrelated errors (Mason

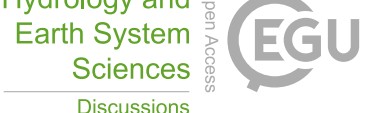



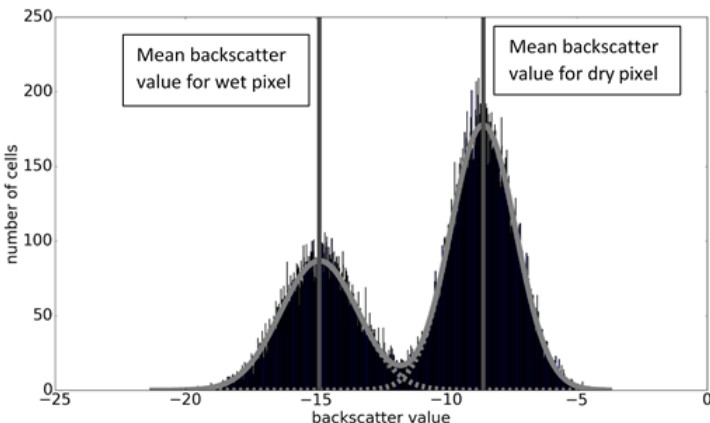

**Figure 4.** Histograms and fitted Gaussian distributions of synthetic backscatter values. Dashed grey lines show two fitted Gaussian distributions and the solid grey line shows the sum of the two fitted distributions. Vertical lines show the positions of the mean wet and dry backscatter values.

et al. (2012)); this allows use of a diagonal observation error covariance matrix. Secondly, without ensemble localisation, using a number of observations larger than the number of ensemble members can cause the assimilation algorithm to overfit the observations (Kepert (2004)).

We have used twice as many observations for the backscatter operator than in the other two cases. The flood edge operators both use the elevation at a flood edge as the flood edge water level observation (defined as the elevation at the first dry pixel). For the backscatter operator, we used two observations for each flood edge: the backscatter value at the first dry pixel and the backscatter value at the last wet pixel. In this way, we provide the same information to each of the observation operators about where the flood edge is. In reality backscatter values in adjacent cells are likely to have correlated observation errors and this is something which requires further research with real backscatter observations.

## 4.5 Observation error covariance matrices

It is important to specify the observation error statistics in data assimilation. In all cases we assume that our observation errors are uncorrelated so that we can use a diagonal error covariance matrix, **R**. We assume that the error in flood edge WLOs is $0.25m$. This is close to the calculated error in SAR-derived water level observations in Mason et al. (2012), and is the same value used in Cooper et al. (2018) and Garcia-Pintado et al. (2015).

The uncertainty in each backscatter observation reflects the distribution to which it belongs (wet or dry). We assume that each entry can be set to be $\sigma_{d1}^2$ corresponding to a dry observation or $\sigma_{w1}^2$ for a wet observation.



## 4.6 Further experimental details

We present here the results from a number of data assimilation experiments, each lasting for a total simulation time of 112 hours. This includes an initial spin-up period with constant inflow for 4 hours (as shown in figure 2) to allow the water to reach an equilibruim state. In each experiment we use 100 forecast ensemble members. Assimilations are carried out at 12 hourly intervals as this is currently the shortest likely return time for satellites equipped with SAR instruments. The ETKF is used without localisation or inflation in all of the experiments as we did not encounter any spurious correlations or problematic ensemble collapse (see Petrie and Dance (2010)). This suggests that 100 ensemble members is sufficient in this particular case.

Experiments were run as follows

– State only estimation. State estimation experiments show how well data assimilation is able to correct forecast water levels at each observation time using the three different observation operators. In all of the experiments, a large bias is present in the forecast channel friction parameter values, which means that by design the error between the ensemble forecast and the truth growns quickly during each forecast step; the forecast corresponding to each of the observation operators relaxes to the same no assimilation (open loop) forecast. This allows us to examine the effect of each observation operator on the water levels in isolation at each observation/assimilation time, as the operators are each acting on very similar pre-assimilation forecasts.

State only estimation experiments were carried out using a positive bias in the forecast channel friction parameter, which leads to forecast water levels that tend to be deeper than the truth (experiment PBSO) at any given cross section, and with a negative bias in the channel friction parameter, leading to shallower forecast water levels (experiment NBSO).

– Joint state and parameter estimation. Updating the value of $n_{ch}$ along with water levels allows us to see the effect of the observation operators on the forecast when the large parameter bias can also be corrected by the assimilation process. Correcting the channel friction parameter in this way leads to better persistence in the forecast correction (Cooper et al. (2018)). Experiments were again carried out using both a positively biased initial channel friction parameter distribution for the forecast ensemble (experiment PBJ) and negatively biased initial channel friction parameter distribution (experiment NBJ).




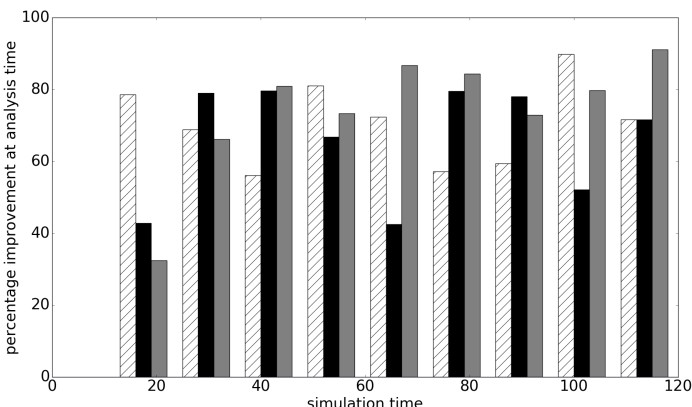

**Figure 5.** Improvement in the forecast at each assimilation time, PBSO experiment. The hatched white bars show improvement for the $\mathbf{h}_s$ operator, the black bars show improvement for the $\mathbf{h}_{np}$ observation operator and the grey bars show the improvement for the $\mathbf{h}_b$ observation operator.

## 5 Results and discussion of update mechanisms

### 5.1 State only estimation

#### 5.1.1 Positive bias in forecast ensemble channel friction parameter (PBSO)

Figure 5 shows improvement in the analysis compared to the forecast at each observation time for the PBSO experiment. Improvement is defined as

$$improvement = \frac{(\overline{\mathbf{x}^f} - \mathbf{x}^t) - (\overline{\mathbf{x}^a} - \mathbf{x}^t)}{\overline{\mathbf{x}^f} - \mathbf{x}^t} \times 100, \qquad (22)$$

where $\mathbf{x}^t$ is the true state of the system. This improvement measure is positive when the error in the analysis is smaller than the error in the forecast, while negative values imply a larger error in the analysis than the forecast. A perfect analysis ($\mathbf{x}^a = \mathbf{x}^t$) would result in a 100% improvement measure. Figure 5 shows that in the PBSO experiment all of the operators reduce the difference between the forecast mean and the truth at each observation time. We found that the error in the forecast then quickly relaxed back to the no assimilation (open loop) case for all of the observation operators. This short lived persistence in forecast improvement (less than approximately 3 hours here) when only water levels are updated is typical for such systems and is reported in many studies, including Cooper et al. (2018), Andreadis et al. (2007), Neal et al. (2009), Garcia-Pintado et al. (2013) and Matgen et al. (2010).





### 5.1.2 Negative bias in forecast ensemble channel friction parameter (NBSO)

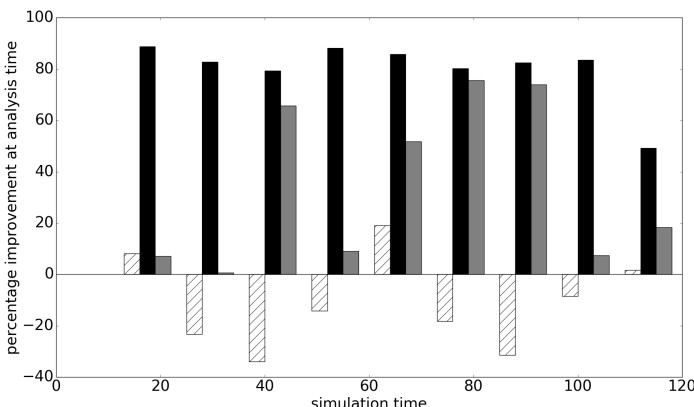

**Figure 6.** Improvement in the forecast at each assimilation time, NBSO experiment. The hatched white bars show improvement for the $\mathbf{h}_s$ operator, the black bars show improvement for the $\mathbf{h}_{np}$ observation operator and the grey bars show improvement for the $\mathbf{h}_b$ observation operator.

Figure 6 shows the improvement in the forecast at each the assimilation time for the NBSO experiment. Here, the ensemble channel friction parameters are such that the mean forecast water level tends to be shallower than the truth at any given cross section in our domain. Unlike in the PBSO experiment, the operators do not all provide a good analysis at every observation

5    time. In fact, assimilation of flood edge observations using the simple flood edge observation operator, $\mathbf{h}_s$, makes the forecast significantly worse at many assimilation times. The reason for this is illustrated by considering the innovation produced by the simple flood edge operator when the forecast is shallower than the truth. The types of innovations produced for mean forecasts that are either deeper or shallower than the truth are shown in a schematic in figure 7.





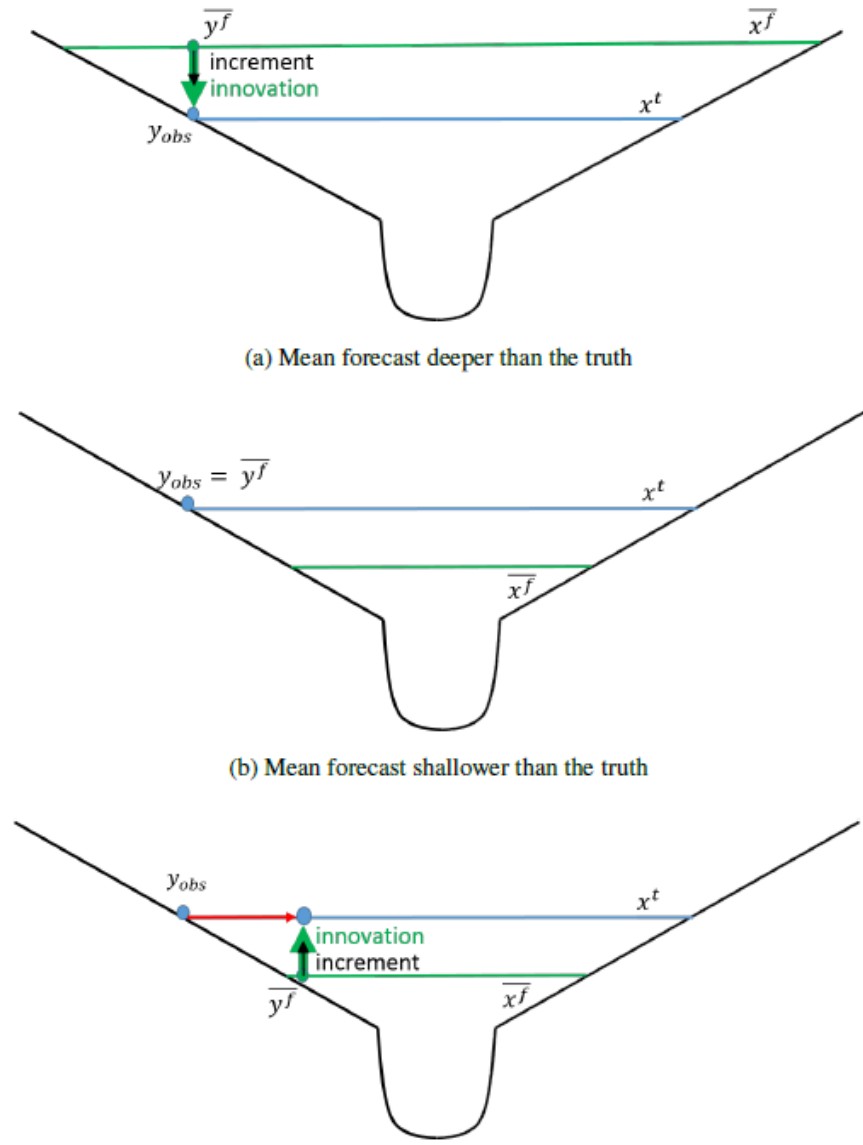

**Figure 7.** Schematic showing innovation for flood edge observation. In all cases blue lines represent the true water level and blue circles represent the corresponding flood edge observation, $y_{obs}$. Green lines show the mean forecast water level and green circles show the corresponding mean forecast-observation equivalent, $\overline{y^f}$. Innovations ($\delta$) are shown with green arrows and increments by thinner black arrows - see equation (11) for definitions.





Figure 7(a) shows a simple domain in a cross section where the mean forecast is deeper than the truth, with the innovation generated by the simple flood edge operator. The innovation is such that the data assimilation algorithm can generate an increment and adjust the forecast water levels to be closer to the true water levels. However, as shown in figure 7(b), when the mean forecast is shallower than the truth, the simple flood edge assimilation method generates an innovation equal to zero.

This is because the observation implies that at the flood edge, the water depth relative to the topography is zero; the ensemble forecast mean also predicts that the water depth is zero at the observation position. The increment is therefore also zero and the forecast cannot be adjusted to be closer to the truth (i.e. to shallower water levels), even though the observation clearly indicates that this is necessary. Figure 7(c) illustrates the way that the nearest wet pixel approach solves this problem by taking the water elevation at the observation position and extrapolating it in space. This effectively moves the observation location to

the nearest wet pixel, allowing a non-zero innovation to be calculated.

Figure 7 illustrates the fact that the simple flood edge operator cannot produce an analysis which is closer to the truth than the forecast when the mean of the forecast ensemble is shallower than the water level predicted by the observation at the observation location. Figure 6 shows that in our experiments the simple flood operator in fact makes the forecast worse, increasing error relative to the truth at several assimilation times. The reason for this is linked to the fact that it is possible for

the mean of the forecast ensemble to be deeper than the truth on the floodplain but shallower than the truth in the river channel.

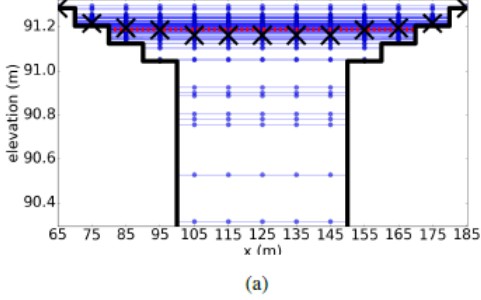

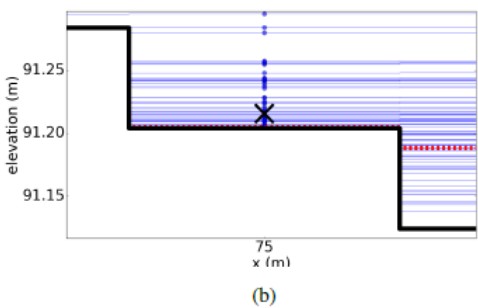

(a)                                                                              (b)

**Figure 8.** Cross section of the domain showing bathymetry as a solid black line. The true water level is shown as a dotted red line, water levels predicted by each ensemble member are shown as blue circles. The mean forecast in each model cell is shown as a cross. Figure 8(a) shows the central part of the domain from $65 \leq x \leq 185$m. Figure 8(b) shows the forecast water levels and resulting forecast mean in the cell centred at 75m in greater detail. Reprinted from Cooper et al. (2018) with permission from Elsevier

Figure 8 shows the domain at one cross section. In figure 8 we see that in the channel (e.g. at $x = 125$m) the true water level is deeper than the ensemble mean. At the edge of the flood, the true water depth is (by definition) zero relative to the topography and the majority of ensemble members also predict zero water depth in these cells. However, there are a small number of ensemble members that predict non-zero water depth at the flood edge; it follows that the ensemble mean at this

location is therefore a small non-zero water depth as per equation (1). The flood edge operator therefore generates an innovation



such that the mean forecast water depth at the flood edge is reduced and the analysed water depths are closer to the truth at this location. Correlations between water levels in the domain mean that the water depth in the channel is also reduced by the update step; this increases the error relative to the truth in the channel. This explains the overall increase in error at assimilation times seen in figure 6.

5    The results in figures 5 and 6 show that the new backscatter operator works well at most of the observation times. The mechanism by which the backscatter observation operator works is illustrated in figure 9.

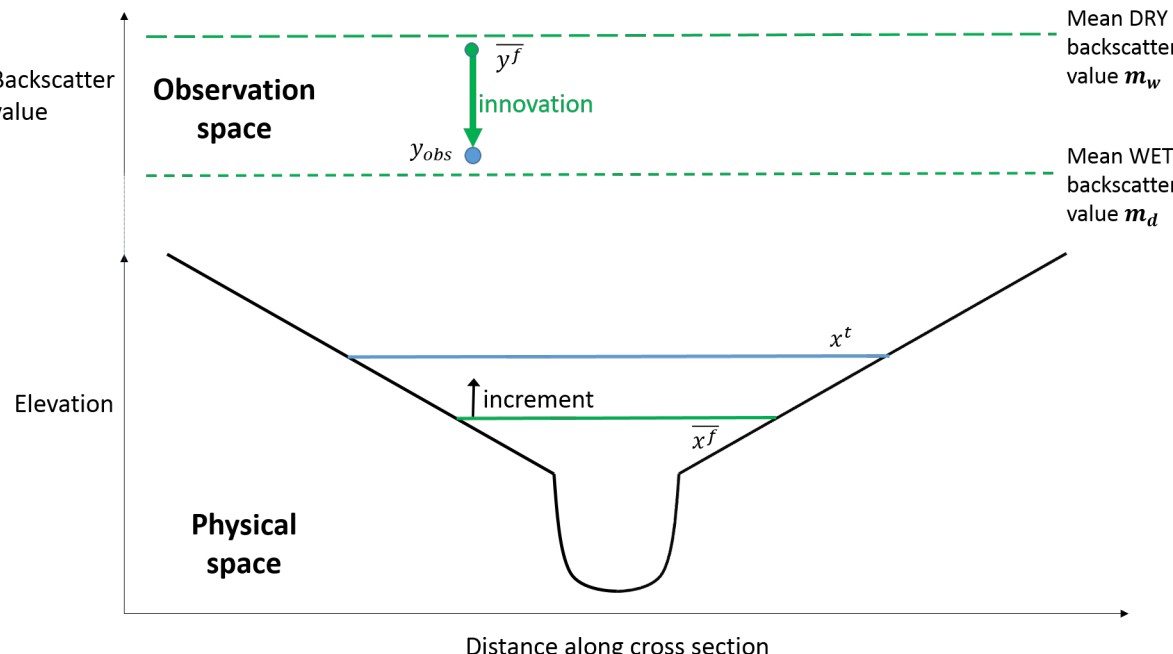

**Figure 9.** Schematic of innovation (in backscatter/observation space) and increment (in physical space) for one backscatter observation. The horizontal blue line represents the true water level and the blue circle represents a corresponding backscatter observation, $y_{obs}$. The solid green line shows the mean forecast water level and the green circle shows the corresponding mean forecast-observation equivalent in observation space, $\overline{y^f}$. The innovation (defined in equation (11)) is shown in observation space with a green arrow and the increment in physical space at the observation position (equation (12)) is represented by a thinner black arrow.

Figure 9 shows a simplified river channel in cross section. The lower part of the figure shows an example of a true and mean forecast water level, as in figure 7. The upper part of the figure shows the same cross section, but is a representation in observation space of an example (single) observation and equivalent mean forecast backscatter value, $\overline{\mathbf{y}^f}$. The green circle in

10   observation space shows $\overline{\mathbf{y}^f}$ in the cell at the observation position. The value of $\overline{\mathbf{y}^f}$ is calculated using water levels forecast by all the ensemble members, through equation (19), and is essentially a measure of the proportion of ensemble members which predict that cell to be wet (or dry). The mean forecast backscatter, $\overline{\mathbf{y}^f}$, will always take a value between the mean observed wet





value, $m_{w1}$ and the mean observed dry value, $m_{d1}$; if half the ensemble members predict a cell to be dry and half predict it to be wet, the value of $\overline{\mathbf{y}^f}$ will lie halfway between $m_{w1}$ and $m_{d1}$. If most ensemble members predict the cell to be wet (dry), the value of $\overline{\mathbf{y}^f}$ will be close to the mean observed wet (dry) backscatter value. The observed backscatter value, $\mathbf{y}_{obs}$, is shown as a blue circle in observation space.

5    The innovation is shown in observation space in figure 9. The innovation is the difference between the observed backscatter value, $\mathbf{y}_{obs}$, and the mean forecast backscatter value, $\overline{\mathbf{y}^f}$. Figure 9 shows that for the $\mathbf{h}_s$ and $\mathbf{h}_{np}$ the state variables and observed variables are the same. In the approach using $\mathbf{h}_b$, the observations are different to the state variables. For $\mathbf{h}_b$ the increment is the calculated difference in water level between the forecast and the analysis in metres, but this is calculated using an innovation that is a difference in backscatter value between the model and the observation. In the example shown, the mean forecast backscatter value indicates that most of the ensemble members predict the cell containing the observation position to be dry. This corresponds to the shallow mean water level prediction shown in physical space. The backscatter observation indicates that the cell is wet. The innovation is therefore large, and indicates that the cell is more likely to be wet than the forecast indicates. This maps into an increment in physical space through equation (12) such that the calculated analysis water level at the observation position is deeper than the forecast water level.

15    A potential problem with the backscatter operator can be illustrated through inspection of equations (13) and (10). Equation (10) shows that when the value of the Kalman gain matrix is zero, there can be no update to the forecast through assimilation of observations, even when there is a large innovation - i.e. a large difference between a model prediction and an observation. Equation (13) shows that this $\mathbf{K} = 0$ condition can be met if either $\mathbf{X} = 0$ or $\mathbf{Y} = 0$. For $\mathbf{Y} = 0$ to be true, it is only required that the ensemble members all predict the cell containing the observation to be dry, or all ensemble members predict the cell to be wet. This is because if all ensemble members predict a cell to be wet then they all give the same value of $\mathbf{y}_i^f = m_w$ through equation (20). Equation (19) then shows that the value of $\overline{\mathbf{y}^f}$ will then also be equal to $m_w$, and each term in $\mathbf{Y}$ must therefore be zero according to equation (9), since all the ensemble members are the same as the mean. This means that if all the ensemble members predict different but positive water depths (i.e. no non-zero water depths are predicted in the ensemble), no increment can be generated and no update made to the forecast, regardless of whether the observation indicates a wet or dry condition.

25    For this reason, observations at or near the edge of the flood are most valuable to the data assimilation algorithm when using the backscatter observation operator, since these are locations where it is most likely that the ensemble members will predict a variety of wet/dry predictions. We did not observe any situation in which $\mathbf{Y} = 0$ in these experiments. It would in principle be possible to add a small amount of noise to each value of $\mathbf{y}_i^f$ in order to prevent $\mathbf{Y} = 0$, but this risks generating an innovation and increment such that the analysis error is larger than the forecast error.

## 5.2    Joint state-parameter estimation

The large source of error in these experiments is, by design, due to a large bias in the forecast ensemble channel friction parameter values. In this section we show the results of updating the forecast channel friction parameter values as part of the assimilation process. One way to measure the effectiveness of a data assimilation approach is to compute the root mean square





error (RMSE) between the resulting forecast and the truth. Here, RMSE is defined as

$$RMSE = \sqrt{\frac{1}{N}\sum_{j=1}^{N}(d_j^t - d_j^f)^2}, \tag{23}$$

where $d_j^t$ is true water depth in the $j$th cell; $d_j^f$ is mean forecast water depth in the same cell. As before, $N$ represents the number of cells in the domain.

**5.2.1    Positive bias in forecast ensemble channel friction parameter (PBJ).**

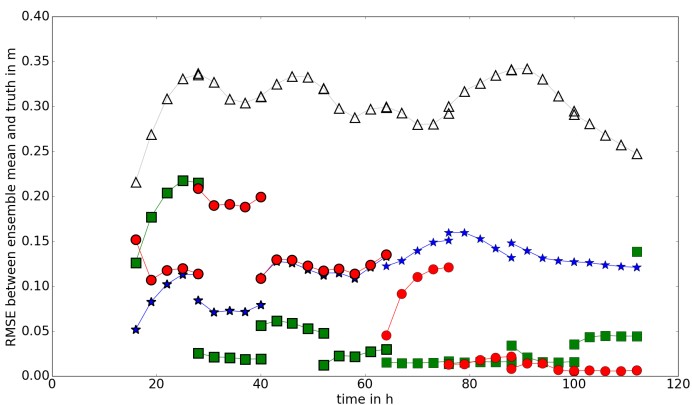

**Figure 10.** RMSE between forecast and truth, PBJ experiment. Open triangles show the RMSE between the open loop forecast and the truth. Blue stars, green squares and red circles show the RMSE between the forecast mean and the truth using the $\mathbf{h}_s$, $\mathbf{h}_{np}$ and $\mathbf{h}_b$ observation operators respectively.

Figure 10 shows the RMSE between the mean water levels predicted by the model and the true water levels for the PBJ experiment. The mean value of $n_{ch}$ and the mean value of the predicted water levels are updated at 12 hourly intervals starting from 16h. At each assimilation time the RMSE for both the forecast (pre-assimilation) and analysis (post-assimilation) water levels are shown; points within a forecast step are joined with a line. The results show that the assimilation leads to a much

improved forecast of water levels for all of the operators at all times. There is persistence in the improvement to the forecast, and each of the observation operators provides a better forecast than the open loop ensemble for the whole of the simulation time. The results obtained using the $\mathbf{h}_s$ operator converge to higher RMSE values than the other two operators. Use of the $\mathbf{h}_b$ operator shows a gradual reduction in RMSE over successive forecast-analysis cycles. The results for the $\mathbf{h}_{np}$ operator show faster reduction in the RMSE values, but the final analysis value (at 112h) has a much higher RMSE. This is because at 112h

the inflow has reduced such that the water is well back within bank and in these conditions the assumptions used to derive water elevation observations break down; the sides of the river are too steep for the water edge position to accurately determine





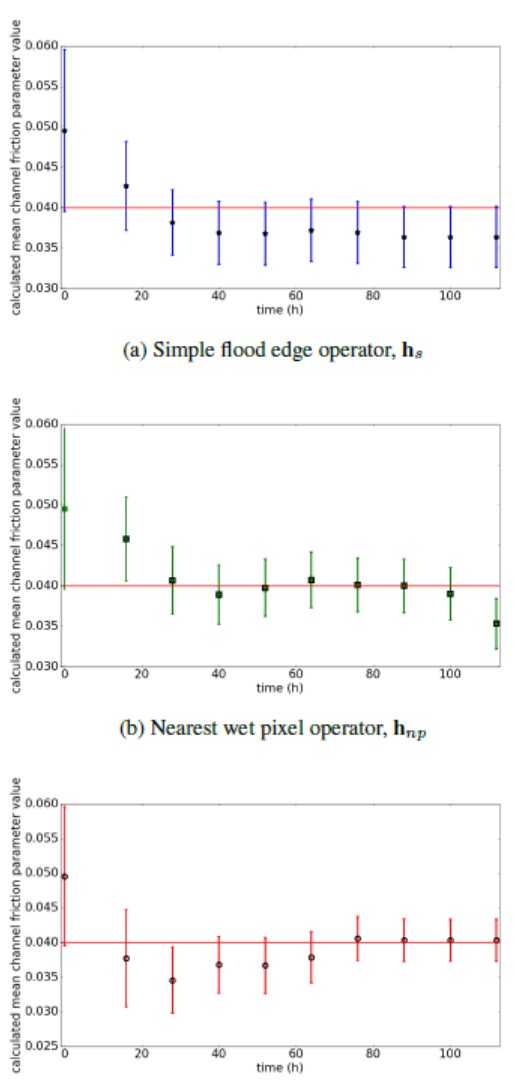

(a) Simple flood edge operator, $\mathbf{h}_s$

(b) Nearest wet pixel operator, $\mathbf{h}_{np}$

(c) Backscatter operator, $\mathbf{h}_b$

**Figure 11.** Calculated analysis mean channel friction parameter, PBJ experiment. Horizontal red line shows true value of channel friction parameter. Error bars show one std of ensemble parameter distribution.

elevation. In an operational setting, it would be necessary to test for an in-bank condition and discard observations for the $\mathbf{h}_{np}$ operator when the river is within bank. This means that it is not possible to calibrate a hydrodynamic model on a river using SAR images when it is not in flood if water level observations are being used (i.e. with either the $\mathbf{h}_s$ or $\mathbf{h}_{np}$ observation operator).





Figure 11 shows the calculated (analysis) mean channel friction parameter values at each assimilation time for the three observation operators. All of the operators produce values for the parameter that are closer to the truth than the initial value. The value of the channel friction parameter calculated using the $\mathbf{h}_b$ observation operator converges to a value close to the truth after 6 observations and then remains there. The value calculated using $\mathbf{h}_{np}$ converges more quickly to a value close to the truth, but the last value in the time series (at 112h) then diverges from the true value. This is because the river is now well within bank and water elevation observations cannot be reliably determined.

### 5.2.2 Negative bias in forecast ensemble channel friction parameter (NBJ)

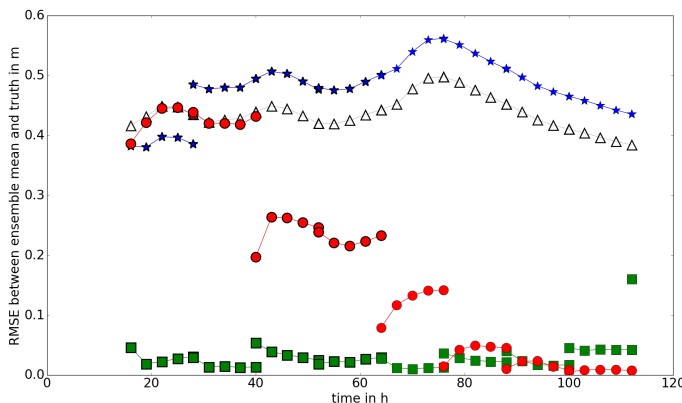

**Figure 12.** RMSE between forecast and truth, NBJ experiment. Open triangles show the RMSE between the open loop forecast and the truth. Blue stars, green squares and red circles show the RMSE between the forecast mean and the truth using the $\mathbf{h}_s$, $\mathbf{h}_{np}$ and $\mathbf{h}_b$ observation operators respectively.

Figure 12 shows the RMSE between the forecast and the truth for the NBJ experiment. The nearest wet pixel approach provides a forecast which is very close to the truth for most of the simulation time. The backscatter operator performs well after the first two assimilation steps, showing a slower convergence to the true solution as in the PBJ experiments. The simple flood edge operator performs badly, leading to a forecast which is worse than the open loop case for most of the time. The reason for the poor performance in this particular experiment is likely due to the mechanisms outlined in section 5.1.2. The forecast is adjusted in the wrong direction at the first assimilation time (at 16h) such that the water levels are too shallow; the mechanism by which this can happen is demonstrated in figure 8. All subsequent corrections are very close to zero, due to the mechanisms illustrated in figure 7, so that the blue line appears to be unbroken.

Figure 13 shows the calculated analysis mean channel friction parameter values at each assimilation time in the NBJ experiment for the three observation operators. The results for the simple flood edge operator support the scenario outlined above, whereby the friction parameter is initially adjusted in the wrong direction and then cannot be updated towards the truth. Al-



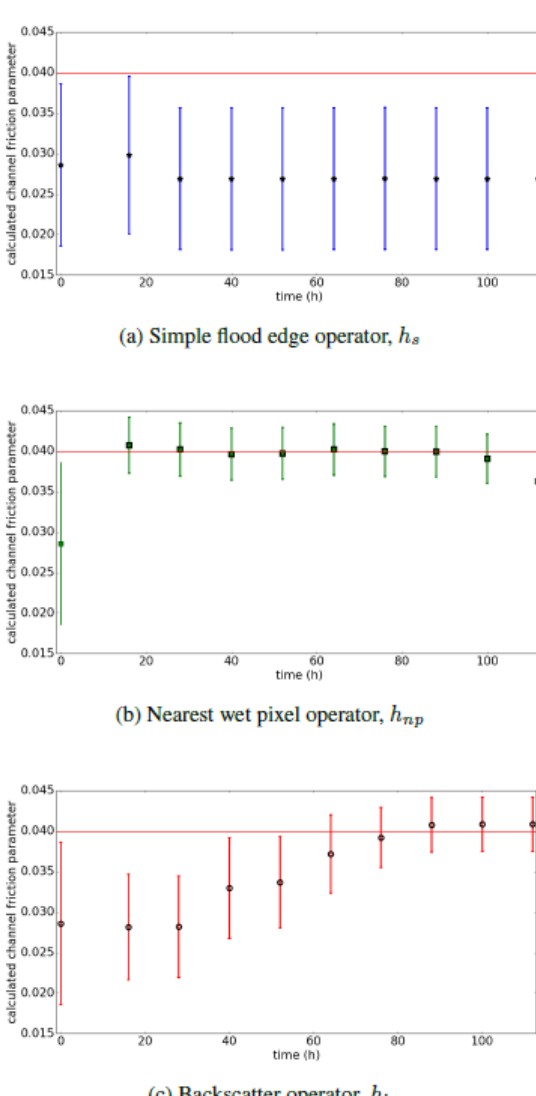

**Figure 13.** Calculated analysis mean channel friction parameter, NBJ experiment. Horizontal red line shows true value of channel friction parameter. Error bars show one std of ensemble parameter distribution.





though the details of this will depend on topography, observation error and choice of forecast inflows and parameters, this is nevertheless an important mechanism to consider when choosing an observation operator. Both the $\mathbf{h}_{np}$ and $\mathbf{h}_b$ operators do successfully correct the value of the parameter towards the truth, with the $\mathbf{h}_{np}$ operator recovering a good value in a shorter time than the $\mathbf{h}_b$ operator. Both figures 12 and 13 show that at the final assimilation time, the analysis and parameter value

provided by the nearest wet pixel operator are not close to the truth. Again, this is because the river is well within bank so the flood edge observation is on ground which is too steep to provide a good observation; in operational settings observations such as these would be screened out and no update would be made with the operator.

## 6    Conclusions

We have carried out a series of experiments to test the performance of three different types of observation operators in an

ETKF approach to data assimilation for fluvial inundation forecasting. Although the results are for one specific idealised domain, one realisation of true inflow and a single realisation of observation error per observation type, we believe that many of our conclusions will be applicable much more widely through the mechanisms we describe. Repeats of experiments (not reported here) with different realisations of observation error show evidence of the same behaviour in terms of the mechanisms we have described. Our experiments show that:

– Simple assimilation of flood edge water elevation observations can result in no correction to the forecast even when there is a large difference between the forecast and the observation. This happens when both the model prediction and the observation predict no flooding at the observation location. We have illustrated the physical mechanism responsible for this (figure 7) and shown an example in which this happens in our experiments (see assimilation times from 28h onwards in figure 12). The simple flood edge operator can also generate an update such that the analysis has a larger

error than the forecast. This can occur when the forecast is deeper than the truth at the observation position, but shallower than the truth in the channel. In such cases the assimilation updates the water levels to shallower levels as required at the observation position, but also wrongly updates the channel water levels to be shallower. The mechanism for this is shown in figure 8; this is responsible for the negative improvement measures in the NBSO experiments (see figure 6). We have shown in our experiments that the simple flood edge operator fails in these ways when the mean ensemble channel

friction parameter is negatively biased but it would also fail if, for example, the mean forecast inflow was negatively biased since errors in friction parameter and inflow are correlated (Cooper et al. (2018)). Since in operational settings both forecast inflow and channel friction parameter values are uncertain, we conclude that the simple flood edge operator is not a good choice.

– The nearest wet pixel approach provides better assimilation accuracy than simple flood edge assimilation: in our experi-

ments we find no evidence of negative 'improvement' scores or zero increments when the forecast and observations are very different. In our idealised system it is the best choice of observation operator in terms of better forecast accuracy in the state only experiments and in terms of rapid convergence to the true solution for both water levels and mean

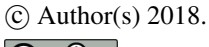



channel friction parameter value in the joint state-parameter experiments. However, we have shown that using water edge observations when the river is well within bank can lead to a degradation of the forecast. Also, locating the nearest wet pixel is likely to be difficult in practise for operational applications using real, more complicated topography. One way to limit the distance between the flood edge observation position and the nearest wet pixel is to locate the nearest

pixel at which some threshold of ensemble members predict a positive water depth. The predicted water elevations at this location could then be used to create $\overline{\mathbf{y}^f}$. This approach balances out the need to include information from ensemble members predicting shallow water levels at the observation position with the requirement that the nearest wet pixel is not too far from the observation location.

– Our new backscatter observation operator performs well compared to more conventional options in our idealised domain

using synthetic observations. The operator does not suffer from the problems of the simple flood edge operator and is able to correct the forecast for the state only assimilation cases. The backscatter operator approach also allowed the forecast to converge to the true solution for both water levels and channel friction parameter value in the joint state-parameter experiments, although in our experiments convergence was slower than for the nearest wet pixel approach. Using backscatter values operationally may speed up the time taken from image acquisition to assimilation and an improved forecast due

to fewer steps in the processing. The new operator could also potentially allow the use of much more information from any given SAR image, although there is likely to be a limit to the number of backscatter observations that can be used without causing variance collapse in the channel friction parameter distribution. Tests using larger numbers of backscatter observations have not been presented here; we plan to address this question in a real case study so that the results will be more directly applicable to real world situations.

This work has shown that our novel backscatter operator has the potential to improve inundation forecasting in fluvial floods, and we believe it may have applications in other types of flooding where SAR images are available. Further work is required to test the operator against the $\mathbf{h}_{np}$ approach in a real case study, using real SAR data and real topography in order to further assess the strengths and weaknesses of the different approaches. We have explained the physical mechanisms associated with the assimilation increments for each type of observation operator; these mechanisms will also be applicable to variational data

assimilation methods using similar observations. Improved understanding of these physical mechanisms provides insight into the best approaches to improve the effectiveness of assimilation of SAR data in the future.

*Code availability.* The inundation simulations in this work were generated using Clawpack 5.2.2, a collection of FORTRAN and python code available from http://www.clawpack.org/ Copyright (c) 19942017, The Clawpack Development Team. All rights reserved. See website for licensing information. Details of the amended Clawpack source code as used in this work are freely available on request from

the corresponding author, as is the python code used to perform data assimilation on the inundation simulation output. Please contact e.s.cooper@pgr.reading.ac.uk for details.

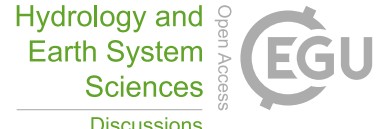

*Author contributions.* ESC ran the experiments and drafted the manuscript. SD, JG-P, NN and PS contributed to analysis of the results, the discussion and manuscript editing.

*Competing interests.* The authors declare that they have no conflict of interest.

*Acknowledgements.* The authors gratefully acknowledge the NERC SCENARIO studentship NE/L002566/1 supporting Elizabeth Cooper,
5  and CASE sponsorship from the Satellite Applications Catapult. This work was also supported in part by NERC grants NE/K00896X/1 and
NE/K008900/1 as well as EPSRC grant EP/P002331/1 and the NERC National Centre for Earth Observation (NCEO).




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
