# Peer review of "Observation operators for assimilation of satellite observations in fluvial inundation forecasting"

_Hydrology and Earth System Sciences, 2018_

## Referee Comment (RC1) · Anonymous Referee #1 · 16 Jan 2019

The manuscript "Observation operators for assimilation of satellite observations in fluvial inundation forecasting" by Cooper et al. aims at improving the methodology for assimilating SAR-derived observations into hydrodynamic models for flood forecast. More specifically, the study investigates the use of three observation operators, which are (1) water level observations at flood edge position, (2) water level observations at the nearest wet pixels, and (3) backscatter measurements. Use (1) and (2) has been previously investigated; the proposed analysis focuses on the use of (3) within an ensemble Kalman filtering approach. Synthetic twin experiments are used to investigate the physical mechanisms by which different observation operators update modelled water levels; the performances of these operators are analysed to compare

the strengths and weaknesses of the three different approaches. The non-linear hydrodynamic model Clawpack is used simulate the propagation of a 110 hours flood hydrograph in an idealised domain. A model realization is considered as truth and it is used to produce synthetic SAR observations. Uncertainties in the input hydrograph and in the assessment of the channel roughness are considered in this study. The ensemble Kalman filtering framework is used to assimilate (1), (2), or (3) within a state updating and a state and parameters updating approach. Assimilations are performed at 5 transects and every 12 hours. The synthetic experiments confirmed the conclusions of previous studies on the limitations of the observation operator (1). This manuscript provides a detailed description of the physical mechanisms leading to the highlighted limitations. Observation operators (2) and (3) were both effective, with (2) allowing quicker retrieval of the truth values but having problems in in-bank flow conditions. Backscatter observation operator (3) also allowed the forecast to converge to the true solution for both water levels and channel friction parameter value in the joint state-parameter experiments.

In my opinion, this study is interesting and provides a valuable contribution to the literature. The manuscript is well written, the introduction is explicative and provides a comprehensive overview of the problem, the methodology is detailed, the results are presented in an effective manner, the discussion provides a thorough analysis, and the conclusions provide a clear summary of the main findings of the study.

I would like to recommend the publication of this manuscript after minor revisions.

More specifically, I would like to suggest some clarification on the methodology and a further discussion on the potential practical use of the observation operator (3).

Firstly, I would like to recommend rewording the sentence in lines 4-9 page 14. The spatial distribution of the backscatter observations is a crucial aspect of this study and I think that this sentence is a bit confusing. The authors might (or might not) consider adding a graphic explanation (maybe by adding details to one of the figures of the

manuscript).

Secondly, I understand that the backscatter observation operator (3) can reduce the computational time and it has the potential to enable the assimilation of a larger number of data. In fact, (3) can be used when a DEM is not available thus allowing SAR data assimilation in a larger number of case studies. Moreover, (3) can enable the assimilation of a larger number of points within the same case study. Do the authors think that the merits of observation operator (3) are due to this "increased data availability" and/or to the different physical mechanism underpinning the assimilation of backscatter rather than SAR-derived water level values? One thing that I could not understand from the manuscript is whether the methodology implied that a pixel with backscatter lower than the mode mw is more likely to be flooded than a pixel having backscatter mw.

As the authors underlined, the number and spatial distribution of backscatter observations used in the data assimilation approach has to be carefully defined. I think that this is a critical aspect for the practical application of the methodology. SAR-derived inundation maps are affected by a large number of uncertainties. For instance, the histogram analysis used here might not be a reliable approach in catchments with emerging flooded vegetation where double bounce effects are not limited to the flood edge. The flood edge itself is often the area of largest uncertainty. Furthermore, SAR-derived inundation maps computed using histogram thresholding are often fragmented and further analysis steps such as region growing or use of ancillary data are required to produce a "continuous" inundation layer. These steps are not included in the methodology presented in the manuscript. Would the authors recommend adding these (or similar) steps in a real case scenario? Pixels within flooded areas might have large backscatter due to double bouncing effects, speckle, and other uncertainties. If such pixels are used as backscatter observations, the data assimilation approach will degrade the performance of the flood forecasting model. Is this a possible scenario? If so, how do the authors recommend avoiding this problem?

I understand that this paper focused on a synthetic experiment and I agree with the authors that a detailed investigation of physical mechanisms within a simplified context is essential to develop knowledge and to explore the feasibility of proposed techniques. However, my main comment concerns the effectiveness of the proposed method in a real world scenario. I think that a further discussion on the potential hurdles and possible solutions for the implementation of observation operator (3) in a real case study would facilitate the reception of the proposed approach and encourage its application. The authors might consider adding a short description of the overall characteristics of the real world scenarios for which they think that their method could provide reliable results. An overall description of the characteristics of the real world scenarios for which the backscatter observation operator is not recommended could also provide useful information to the readers.

Finally, I listed below a few detailed remarks.

I hope the authors will find my comments and questions useful to improve their manuscript.

Page 9, line 21: I think the second parenthesis after George (2018) should be removed.

Page 10, line 18: a full stop should be added before "Inflows".

Page 11, line 19: "ch" in the symbol for channel roughness should be a subscript.

Page 12, line 3: are the intervals of roughness values correct?

Page 13, Figure 3: are the values in the x-axis correct?

Page 13, Figure 3: low backscatter values are usually represented with dark grey to black, high backscatter values are usually represented with light grey to white. In the colour scale used in this figure, the higher the backscatter value, the darker the pixel. Despite this is just a cosmetic detail, I was wondering whether the authors are willing to reverse the current colour scale to allow a more straightforward interpretation of the figure.

Page 15, line 12: "growns" should be corrected.

Page 18, Figure 7c: the authors might consider adding a description of the red arrow.

Page 19, line 12: the authors might consider rewording the sentence "the water level predicted by the observation at the observation location" to improve the readability of the paragraph. More specifically, "predicted" might not be the most appropriate word in this context.

---

## Referee Comment (RC2) · Anonymous Referee #2 · 17 Jan 2019

The paper compares three observation operators in the framework on Ensemble Kalman filtering when SAR observations are assimilated in a hydrodynamics model. This is very interesting piece of work and clearly deserves to be published in HESS as it deals with the use of innovative and rich data and aims at correcting model parameters and state to improve flood forecasts. The paper is very well written, the state of the art well described and the results presented in a clear manner in the context of synthetic experiments when the control vector is defined as the state vector or augmented with the model friction coefficient. However, a number of issues should be address before the paper can be considered for publication. Some details on the data assimilation algorithm, especially the cycling should be added and some clarification

on the observation operators should be given. The major comments are listed below.

Introduction

- References from Oubanas should be included in the state of the art when referring to hydrodynamics data assimilation, with remote sensing data.

Oubanas, H. : Variational assimilation of satellite data into a full saint-venant based hydraulic model in the context of ungauged basins. Ph.D. thesis, Institut National des Sciences Appliquées de Toulouse (2018)

Oubanas, H., Gejadze, I., Malaterre, P.O., Durand, M., Wei, R., de Moraes Frasson, R.P., Domeneghetti, A. : Discharge estimation in ungauged basins through variational data assimilation : The potential of the swot mission. Journal of Hydrology pp. 2405–2423 (2018)

Oubanas, H., Gejadze, I., Malaterre, P.O., Mercier, F. : River discharge estimation using variational data assimilation involving the full saint-venant model and synthetic swot-type observations. Journal of Hydrology pp. 405–430 (2017)

- Front like observations were assimilated in the framework of wildfire propagation. Coordinates (x- and y-) of along edge markers were used as observation, avoiding the non-gaussian issue of binary data (burn or unburned area, dry or wet area). This work should be cited in the references as it proposes an alternative to the 3 options presented here.

Rochoux, M. : Vers une meilleure prévision de la propagation d'incendies de forêt : évaluation de modèles et assimilation de données. Ph.D. thesis, Ecole Centrale Paris (2014)

Rochoux, M., Collin, A., Zhang, C., Trouvé, A., Lucor, D. : Front shape similarity measure for front position sensitivity analysis and data assimilation. ESAIM : Proceedings and Surveys 63, pp. 215-236.

Rochoux, M., Ricci, S., Lucor, D., Cuenot, B., Trouvé, A. : Towards predictive data-driven simulations of wildfire spread - part 1 : Reduced-cost ensemble kalman filter based on polynomial chaos surrogate model for parameter estimation. Nat. Hazards and Earth Syst. Sci. 14(11), 2951–2973 (2014)

Part 2

- ETKF algorithm should be presented in more details even though this is a classical algorithm and references are given. The choice of the perturbation matrix is essential in this deterministic filtering algorithm and for the present paper to be self-dependent, a short description of how this is done should be included. - Please justify why using a deterministic filter ETKF instead of a stochastic EnKF?

Part 3

- Part 3.1 should include Figures 7 and 9. - Assuming that the water level is constant perpendicularly to the flow is essential for the second observation operator hnp. While this was mentioned at the end of 3.2 relating to other published papers, this should be mentioned earlier when presenting the operator along with the related difficulties (such as finding the nearest point). - For the backscatter approach, operator hb associates either md or mw to the model equivalent at the observation point. This means that operator hb only returns 2 possible values that are compared to the entire range of backscatter values. As a consequence, any wet pixel in the model state (for instance different WLO values for different members) would return the same equivalent, and the difference between members is lost. I feel, we are losing information in the ensemble here. Yet, I may be missing a point here, so please clarify.

Part 4

- While I am not questioning the use of the Clawpack model in this work, I am curious to know why the authors did not use a community model such as LISFLOOD, MIKE or TELEMAC. Some details on Clawpack model may be included here again for the paper to be a little more self-dependent: is it a full 2D model? (is the water level constant perpendicular to the flow direction ?) How are the limits of the simulation domain prescribed? (solid boundaries?) - The ensemble construction relies on the perturbation of a true inflow with additive time correlated signal, assuming that the correlation length is large. Why not simply using a scalar additive perturbation constant over time, as it basically comes down to the same result? - Cycling of the analysis requires explanations on how the friction coefficient is updated along the analysis. First, it is not clear to me whether the analysis is carried out at an instant of observation or over an assimilation window (like a smoother would be). Secondly, it is mentioned that the friction coefficient is drawn from a normal distribution with mean different from the true friction coefficient. But how is the analysed value of the friction used for the following cycle? Is the friction drawn from a normal distribution with a mean equal to the mean analysis? How about the standard deviation? Is there any inflation on the model parameter to avoid ensemble collapse? Is the corrected value of the friction kept persistent for the forecast? I suggest adding a scheme to properly explain the ensemble cycling in part 4.3. - In the synthetical observations part 4.4, I understand that given the WLO in a flood edge pixel, a backscatter value is drawn from a normal distribution centred in md or mw. Why bother computing the Gaussian fit and new Gaussian values when these observations are going to be compared to binary values (equivalent model state values that are either md or mw)? - The location of the observation is not clear to me: in 4.4, it is said that the flood edge is defined to be the elevation at the first 'dry' pixel encountered when moving in a perpendicular direction from the centre of the channel along one of the defined cross sections then it is said that two observations per flood edge are considered. Please clarify and locate the observations in Fig 3. - I couldn't find information of observation frequency while it is mentioned that the assimilations are carried out every 12 hours. This goes back to my previous question on instantaneous or time window assimilation.

Part 5

- I suggest adding rank diagram to check the validity of the ensemble with regards to the observation. This would be a starting point to identify cases where all members WLO are lower than observation as this case leads to a problematic zero correction in the analysis. This would allow for correcting the ensemble (and its spread) beforehand applying data assimilation while being aware of a problem. - The computation of the ensemble mean at the flood edge illustrated in Fig 8 causes a negative effect of the analysis because the members that are shallower than the observation are associated with a zero WLO at the flood edge, thus not contributing to the mean computation. I regret that no solution was proposed in the paper. A suggestion would be to compute the mean WLO at the centre of the river and assume it is constant perpendicularly to the flow. This assumption is made already for the second operator solution. - I have doubts about Figure 9: the observation is located at the true flood edge, where the innovation is computed. I guess the arrow in the observation space should be translated on the left, above the flood edge. Plus, md and mw are mixed in the right hand side legend. - The results for hb are satisfying while difficulties occur when all members are shallower than the truth or reversely. I regret the authors did not propose an alternative to this while being aware of it. I suppose that in a real case scenario, this situation may occur depending on how the ensemble is generated. Thus, I suggest adding ensemble validity check as well as reconsidering the computation of the model equivalent that binaiyly returns md or mw indistinctly of the water level value.

Part 6

Locating the flood edge seems a difficult task for a real case with a randomly shaped flood surface, also, locating the nearest wet pixel is a complex task in non-idealized cases. I suggest to investigate this topic that is central to all 3 observation operator proposed here.

---

## Author Comment (AC1) · 11 Feb 2019

The authors thank the reviewer for their useful comments, which will help improve the manuscript. Please see the attached supplement for detailed responses to each of the reviewer's comments.

Best wishes, Elizabeth Cooper.

Please also note the supplement to this comment:
https://www.hydrol-earth-syst-sci-discuss.net/hess-2018-589/hess-2018-589-AC1-supplement.pdf

---

## Author Comment (AC2) · 11 Feb 2019

al.

**Elizabeth S. Cooper et al.**

e.s.cooper@pgr.reading.ac.uk

The authors thank the reviewer for their useful comments, which will help improve the manuscript. Please see the attached supplement for responses to the reviewer's comments.

Best wishes, Elizabeth Cooper.

Please also note the supplement to this comment:
https://www.hydrol-earth-syst-sci-discuss.net/hess-2018-589/hess-2018-589-AC2-supplement.pdf

---

## Author Response (AR1)

**Title: Observation operators for assimilation of satellite observations in fluvial inundation forecasting**

**Authors: E. S. Cooper, S. L. Dance, J. García-Pintado, N. K. Nichols and P. J. Smith**

The authors thank the reviewers for their useful comments, which have led to improvements in the manuscript. Below we give each comment in bold (abridged where appropriate) and describe how we have altered the manuscript to address the reviewer's concern. We give changes to the manuscript in italic font. For convenience all alterations in the revised manuscript appear in red/blue.

**Response to reviewer 1:**

1. **I would like to suggest some clarification on the methodology and a further discussion on the potential practical use of the observation operator (3).**
   We have made clarifications to the methodology as described in our responses to the reviewer's specific comments 2 and 4. We have added a 'Discussion' (p. 27, section 6) to the paper to address the practical application of the new observation operator.

   *6. Discussion*

   *'In this study we have chosen to use a small number of backscatter observations for our experiments. This allowed us to compare updates between the three observation operators when the observation operators were all given equivalent information; in this way we can draw conclusions about the physical mechanisms responsible for the different updates. In a real case, one of the major advantages of using our new backscatter observation operator is that it would be possible to use a large number of backscatter observations compared to the number of water level observations that are typically available. The availability of a large number of observations may be a major strength of our new approach; in our simple experiments (not shown) we found that assimilating a larger number of observations with the backscatter operator provided a better analysis than using only a few. Another merit of the backscatter operator is that there is less processing involved in using backscatter observations directly, potentially reducing the amount of time between acquisition of a SAR image and its use to update an inundation forecast. The backscatter operator also removes the need for locating the 'nearest wet pixel' in the model forecast, which can be computationally costly.*

   *There are a number of potential problems with practical implementation of the backscatter operator. One is that using histograms to produce SAR-derived inundation maps can lead to errors in assigning pixels to wet/dry categories. One way to deal with this would be to use region growing techniques (see e.g. Horrit et al (2001)) or change detection techniques (see e.g. Hostache et al (2012)) to produce robust wet/dry maps for SAR images, and then perform a quality control procedure to discard any backscatter observations that would lead to mis-classification due to e.g. emergent vegetation. This procedure would remove the advantage of fewer processing steps for the backscatter operator, but may not be necessary. Further research is required to understand how robust the method is to the proportion of misclassified SAR pixels in a real case study. We note that the backscatter operator would not generate an update to the forecast in model cells that all the ensemble members predicted to be dry (or wet) as discussed in the last paragraph of section 5.1.2.*

*This means that SAR pixels far from the river wrongly classified as wet, or SAR pixels in the river channel wrongly classified as dry would not degrade the forecast through an erroneous update.*

*The new backscatter operator is likely to work well in cases where good separation of the wet/dry distributions can be obtained through a histogram, and less well in cases where the distributions overlap. The new observation operator does not require a digital elevation model to generate forecast-observation equivalents, although the hydrodynamic model would require topography information to generate a forecast. Water level observations cannot be accurately determined in areas with high slope, whereas backscatter observations will be unaffected. Like the other observation operators, the new operator will likely provide better results in rural settings than urban settings; double-bounce and layover effects due to buildings are potential sources of problems for all of the operators (Mason et al. (2018)).*

*Reference: Mason, D. C., Dance, S. L., Vetra-Carvalho, S. and Cloke, H. L. (2018) Robust algorithm for detecting floodwater in urban areas using Synthetic Aperture Radar images. Journal of Applied Remote Sensing, 12 (4). 045011. ISSN 1931-3195 doi: https://doi.org/10.1117/1.JRS.12.045011*

2. **'Firstly, I would like to recommend rewording the sentence in lines 4-9 page 14. The spatial distribution of the backscatter observations is a crucial aspect of this study and I think that this sentence is a bit confusing. The authors might (or might not) consider adding a graphic explanation (maybe by adding details to one of the figures of the manuscript).**
We have added a schematic (Figure 5) to make the observation locations relative to the flood edge clearer and reworded the text (now p14 from line 16):

*'In this study we wish to investigate the differences in the updates generated by different observation operator approaches. We therefore use equivalent observation information for each of the operators. In the case of the water level observation operators we have used flood edge water level observations at six locations, where the flood edge location is defined as the position of the first dry model cell (see section 4.4). For the new operator we use two backscatter observations for each transect.*

*Figure 5 shows a schematic of the locations of the observations we have used in this study, relative to the edge of the flood. All observations used in this study come from transects at y=500m, 700m, 900m, 1100m, 1300m and 1500m. In practical application of the backscatter operator, observations could be used from any location covered by the SAR image.'*

[Figure]

*Caption: Figure 5: Schematic of observation locations used in this study for each transect in cross section. The thick black line shows the discretised domain elevation, the dashed blue line shows the observed flood water level. The arrows and green crosses show the locations of the observations.*

3. **Do the authors think that the merits of observation operator (3) are due to "increased data availability" and/or to the different physical mechanism underpinning the assimilation of backscatter rather than SAR-derived water level values?**

   We have addressed this question in the new 'Discussion' section – see first paragraph of response to comment 1.

4. **One thing that I could not understand from the manuscript is whether the methodology implied that a pixel with backscatter lower than the mode mw is more likely to be flooded than a pixel having backscatter mw.**

   The histograms give us information about the distribution of backscatter values for wet and dry pixels. The two distributions represent the probability that a pixel has a backscatter value, b, given that the pixel is wet, i.e. p(b|w), and the probability that a pixel has a backscatter value, b, given that the pixel is dry, i.e. p(b|d). We do not compute the probability distribution that the reviewer asked about, i.e. the probability that a pixel is wet(dry) given its backscatter value, p(w|b), (or p(d|b)) in this method.  We have added text to clarify this at p9, line 11:

   *'These distributions represent the probability that a pixel has a particular backscatter value, given that the pixel is wet (or dry).'*

5. **As the authors underlined, the number and spatial distribution of backscatter observations used in the data assimilation approach has to be carefully defined. I think that this is a critical aspect for**

**the practical application of the methodology. SAR-derived inundation maps are affected by a large number of uncertainties. For instance, the histogram analysis used here might not be a reliable approach in catchments with emerging flooded vegetation where double bounce effects are not limited to the flood edge. The flood edge itself is often the area of largest uncertainty. Furthermore, SAR-derived inundation maps computed using histogram thresholding are often fragmented and further analysis steps such as region growing or use of ancillary data are required to produce a "continuous" inundation layer. These steps are not included in the methodology presented in the manuscript. Would the authors recommend adding these (or similar) steps in a real case scenario?**

These questions have been addressed in the new 'Discussion' section – see response to comment 1.

6. **Pixels within flooded areas might have large backscatter due to double bouncing effects, speckle, and other uncertainties. If such pixels are used as backscatter observations, the data assimilation approach will degrade the performance of the flood forecasting model. Is this a possible scenario? If so, how do the authors recommend avoiding this problem?**

These points have been addressed in the new 'Discussion' section – see response to comment 1.

7. **I understand that this paper focused on a synthetic experiment and I agree with the authors that a detailed investigation of physical mechanisms within a simplified context is essential to develop knowledge and to explore the feasibility of proposed techniques. However, my main comment concerns the effectiveness of the proposed method in a real world scenario. I think that a further discussion on the potential hurdles and possible solutions for the implementation of observation operator (3) in a real case study would facilitate the reception of the proposed approach and encourage its application. The authors might consider adding a short description of the overall characteristics of the real world scenarios for which they think that their method could provide reliable results. An overall description of the characteristics of the real world scenarios for which the backscatter observation operator is not recommended could also provide useful information to the readers.**

We now address issues relating to practical application of our new backscatter operator in the new 'Discussion' section – see response to comment 1. The authors plan to publish results from applying the new observation operator to a real world case study in a separate paper.

8. **Page 9, line 21: I think the second parenthesis after George (2018) should be removed.**

This has been removed.

9. **Page 10, line 18: a full stop should be added before "Inflows".**

This has been added.

10. **Page 11, line 19: "ch" in the symbol for channel roughness should be a subscript**

This has been corrected.

11. **Page 12, line 3: are the intervals of roughness values correct?**

We have clarified the intervals for the channel friction parameter distributions from p12, line 8:

*'For the initial forecast step, a value of $n_{ch}$ for each forecast ensemble member was drawn from a normal distribution with mean, μ, that is different to the true value and standard deviation σ. ...... Initial forecast channel friction parameters are randomly drawn from a normal distribution with μ= 0.05 and σ = 0.01 for experiments with positive bias in $n_{ch}$ and with μ = 0.03 and σ = 0.01 for experiments with negative bias in $n_{ch}$.'*

12. **Page 13, Figure 3: are the values in the x-axis correct?**
   The values have been corrected.

13. **Page 13, Figure 3: low backscatter values are usually represented with dark grey to black, high backscatter values are usually represented with light grey to white. In the colour scale used in this figure, the higher the backscatter value, the darker the pixel. Despite this is just a cosmetic detail, I was wondering whether the authors are willing to reverse the current colour scale to allow a more straightforward interpretation of the figure.**
   The colour scale on figure 3 has been corrected to match real SAR images as suggested.

14. **Page 15, line 12: "growns" should be corrected.**
   This has been corrected.

15. **Page 18, Figure 7c: the authors might consider adding a description of the red arrow.**
   We have added the following text to the caption (now figure 8, page 19):
   *'The red arrow shows the difference between the observation location and the nearest wet pixel location.'*

16. **Page 19, line 12: the authors might consider rewording the sentence "the water level predicted by the observation at the observation location" to improve the readability of the paragraph. More specifically, "predicted" might not be the most appropriate word in this context.**
   We have replaced this sentence (p20, line 11) with:
   *'Figure 8 illustrates the fact that the simple flood edge operator cannot produce a useful update when the mean of the forecast ensemble is shallower than the observed water level.'*

**Response to reviewer 2:**

1. **References from Oubanas should be included in the state of the art when referring to hydrodynamics data assimilation, with remote sensing data.**
   The authors thank the reviewer for pointing out these papers and have added these references to section 1 of the manuscript, p1 line 22.

2. **Front like observations were assimilated in the framework of wildfire propagation. Coordinates (x- and y-) of along edge markers were used as observation, avoiding the non-gaussian issue of binary**

**data (burn or unburned area, dry or wet area). This work should be cited in the references as it proposes an alternative to the 3 options presented here**

We have added the references as suggested to section 3.3, p10, line 1.

3. **ETKF algorithm should be presented in more details even though this is a classical algorithm and references are given. The choice of the perturbation matrix is essential in this deterministic filtering algorithm and for the present paper to be self-dependent, a short description of how this is done should be included.**

We have added further details of the matrix used to update the perturbations from p6, line 3:

*'The perturbation matrix is updated by the matrix $T \in R^{M \times M}$. We use an unbiased, symmetric square root formulation of the matrix $T$, constructed in a way that ensures that the analysis state error covariance, $P^a = X^a (X^a)^T$ is the same as the analysis error covariance calculated in the Kalman covariance update (in e.g. Kalman (1960)). The formulation makes use of a singular value decomposition (Golub and Van Loan, 1996),*

*$(R^{-1/2}Y^f)^T = U\Sigma V^T$,*

*where $U \in R^{M \times M}$ and $V \in R^{p \times p}$ are orthogonal. The columns of $U$ and $V$ are the left and right singular vectors of $(R^{-1/2}Y^f)^T$ respectively. The diagonal elements of the matrix $\Sigma \in R^{M \times p}$ are the singular values of $(R^{-1/2}Y^f)^T$. A solution for $T$ is then*

*$T = U(I + \Sigma\Sigma^T)^{-1/2}U$,*

*where $I$ is the identity matrix. See Livings et al. (2008), Cooper et al. (2018) for further details of how $T$ is computed.'*

4. **Please justify why using a deterministic filter ETKF instead of a stochastic EnKF?**

The authors follow the approach of Garcia-Pintado et al. (2013), Garcia-Pintado et al. (2015) and Cooper et al. (2018) in using an ETKF for a similar application. We have added text at p3, line 23:

*'....following the approach of Garcia-Pintado et al. (2013), Garcia-Pintado et al. (2015) and Cooper et al. (2018)'*

5. **Part 3.1 should include Figures 7 and 9.**

These figures (now 8 and 10) contain information relevant to the Results and Discussion section which is not relevant to section 3. Nevertheless we have added text in section 3.2 (p8, line 17) to cross-reference the section containing figures 8 and 10:

*'More information about how the observation operator works in a synthetic case is given in section 5.1.2. '*

6. **Assuming that the water level is constant perpendicularly to the flow is essential for the second observation operator hnp. While this was mentioned at the end of 3.2 relating to other published**

**papers, this should be mentioned earlier when presenting the operator along with the related difficulties (such as finding the nearest point).**

In our view it is important to describe the operator in section 3.2 before discussing problems with its practical implementation (these are already discussed later within section 3.2).

7. **For the backscatter approach, operator $h_b$ associates either $m_d$ or $m_w$ to the model equivalent at the observation point. This means that operator $h_b$ only returns 2 possible values that are compared to the entire range of backscatter values. As a consequence, any wet pixel in the model state (for instance different WLO values for different members) would return the same equivalent, and the difference between members is lost. I feel, we are losing information in the ensemble here. Yet, I may be missing a point here, so please clarify.**

It is indeed a feature of the binary backscatter observation operator that ensemble predictions of water depth are converted only to predictions of wet or dry; this is because they correspond to backscatter observations which contain no information about water depths. This is mentioned in the original text as a potential drawback (now p. 22 lines 15-29); in situations where all the ensemble members agree a cell is wet (or dry) no update can be generated even when the observed water level is different to the mean forecast level. However, as mentioned in the new Discussion section (see response to reviewer 1, comment 1) this feature is potentially beneficial as it means the method is robust to outliers and will not update the forecast when, for example, pixels very far from the river are wrongly classified as wet based on backscatter value.

8. **While I am not questioning the use of the Clawpack model in this work, I am curious to know why the authors did not use a community model such as LISFLOOD, MIKE or TELEMAC.**

We chose Clawpack as the code is open source, available for Linux, and uses robust, accurate and efficient numerical solvers which are able to deal effectively with shocks (i.e hydraulic jumps) in the solution. In addition, this work builds on our previously published study Cooper et al (2018).

9. **Some details on Clawpack model may be included here again for the paper to be a little more self-dependent: is it a full 2D model? (is the water level constant perpendicular to the flow direction ?)**

We stated in section 4.1 that the 2D shallow water equations are solved. We have added at p10, line 8:

*'everywhere in the domain'*

10. **How are the limits of the simulation domain prescribed? (solid boundaries?)**

We have added the following to section 4.1: p10, line 11:

*'In our simulations the boundary condition is extrapolating (outflow) on the y=0 boundary; all other boundaries are solid wall.'*

11. **The ensemble construction relies on the perturbation of a true inflow with additive time correlated signal, assuming that the correlation length is large. Why not simply using a scalar additive perturbation constant over time, as it basically comes down to the same result?**

We agree that this would give a similar result; we chose the approach used here, in which perturbations depend on the flow, based on a similar method used to generate inflow ensembles in Garcia-Pintado et al. (2013), Garcia-Pintado et al. (2015) and Cooper et al. (2018).

12. **Cycling of the analysis requires explanations on how the friction coefficient is updated along the analysis. First, it is not clear to me whether the analysis is carried out at an instant of observation or over an assimilation window (like a smoother would be).**
We have added the following sentence on p4 line 14 to clarify:
*'We use the ETKF in its standard application as a sequential filter. As such we perform an update step at the time of each available observation.'*

13. **Secondly, it is mentioned that the friction coefficient is drawn from a normal distribution with mean different from the true friction coefficient. But how is the analysed value of the friction used for the following cycle?**
In the original manuscript we stated that the friction parameter follows the same update-forecast cycle on p6 line 24: 'The augmented state vector is updated by the ETKF algorithm through equations (10) and (14). Parameter value(s) are updated according to the observations due to covariances between errors in the model state and errors in the parameter(s).' We have changed the text from p12, line 8 to make this clearer:
*'For the initial forecast step, a value of $n_{ch}$ for each forecast ensemble member was drawn from a normal distribution with mean, µ, that is different to the true value and standard deviation σ. This imposed bias in the forecast ensemble channel friction parameter means that we can test how well data assimilation with different observation operators can correct the forecast state and parameter value towards the truth. In our state estimation experiments, the value of $n_{ch}$ assigned to each ensemble member remained constant throughout the simulation. For joint state-parameter experiments, the values of $n_{ch}$ were updated at each assimilation time through the ETKF equations, as described in section 2.2. Using an incorrectly specified channel friction parameter in the forecast is realistic, as the true value is unlikely to be known in operational situations. Initial forecast channel friction parameters are randomly drawn.....'*

14. **Is the friction drawn from a normal distribution with a mean equal to the mean analysis? How about the standard deviation?**
We have made this clearer in the text - see response to comment 13. The initial values for the ensemble parameter values are drawn from a Gaussian distribution – see response to reviewer 1, comment 11, for clarification of the distribution characteristics. For analysis values, the perturbation matrix for the augmented case includes the friction parameter perturbations; this follows from equations (17) and (2). The friction parameter perturbations (and therefore the standard deviation of the parameter distribution) are therefore updated through equation (14) in the same way as for the state perturbations.

15. **Is there any inflation on the model parameter to avoid ensemble collapse?**

In the original manuscript we stated on p16 line 16 that we have not used any inflation; this applies to both parameter and state perturbations. As in *Cooper et al. (2018)* we did not observe ensemble collapse in this simple system.

16. **Is the corrected value of the friction kept persistent for the forecast?**
    Equation (18) in the original manuscript showed that the friction values are not changed during the forecast step.

17. **I suggest adding a scheme to properly explain the ensemble cycling in part 4.3.**
    The standard ETKF scheme is used for the ensemble cycling. Our responses to comments 12 – 16 clarify this.

18. **In the synthetic observations part 4.4, I understand that given the WLO in a flood edge pixel, a backscatter value is drawn from a normal distribution centred in md or mw. Why bother computing the Gaussian fit and new Gaussian values when these observations are going to be compared to binary values (equivalent model state values that are either md or mw)?**
    We use the variance of the distributions to provide information about observation uncertainty. This is stated in section 4.5 of the original manuscript.

19. **The location of the observation is not clear to me: in 4.4, it is said that the flood edge is defined to be the elevation at the first 'dry' pixel encountered when moving in a perpendicular direction from the centre of the channel along one of the defined cross sections then it is said that two observations per flood edge are considered. Please clarify and locate the observations in Fig 3.**
    Reviewer 1 made the same point. We have added an extra schematic (figure 5) to make the observation locations relative to the flood edge clearer and reworded the text on p14 from line 16 – see response to reviewer 1, comment 2.

20. **I couldn't find information of observation frequency while it is mentioned that the assimilations are carried out every 12 hours. This goes back to my previous question on instantaneous or time window assimilation.**
    The observation frequency is the same as the assimilation frequency. We have clarified this at p16 line 14:
    *'Assimilations are carried out at 12 hourly intervals. This is currently the shortest likely time between observations due to return times for satellites equipped with SAR instruments.'*

21. **I suggest adding rank diagram to check the validity of the ensemble with regards to the observation. This would be a starting point to identify cases where all members WLO are lower than observation as this case leads to a problematic zero correction in the analysis. This would allow for correcting the ensemble (and its spread) beforehand applying data assimilation while being aware of a problem.**
    For a binary operator a rank histogram in observation space would not give meaningful information, as the value of the forecast-observation equivalent for each ensemble member can only have two

values (wet or dry). We agree that it would be useful to have a method of checking for filter divergence so that the user can check the observations and model forecasts. The best approach to this would need to be determined by experience with real case studies but this is not within the scope of this study.

22. **The computation of the ensemble mean at the flood edge illustrated in Fig 8 causes a negative effect of the analysis because the members that are shallower than the observation are associated with a zero WLO at the flood edge, thus not contributing to the mean computation. I regret that no solution was proposed in the paper. A suggestion would be to compute the mean WLO at the centre of the river and assume it is constant perpendicularly to the flow. This assumption is made already for the second operator solution**
We agree that the flood edge operator gives poor results; we consider the nearest wet pixel approach already discussed in section 3.2 to be a solution to this problem. This improved version effectively uses the approach suggested by the reviewer.

23. **I have doubts about Figure 9: the observation is located at the true flood edge, where the innovation is computed. I guess the arrow in the observation space should be translated on the left, above the flood edge. Plus, md and mw are mixed in the right hand side legend**
We have corrected the $m_d$ and $m_w$ labels and moved the arrow to the left (now figure 10).

24. **The results for hb are satisfying while difficulties occur when all members are shallower than the truth or reversely. I regret the authors did not propose an alternative to this while being aware of it. I suppose that in a real case scenario, this situation may occur depending on how the ensemble is generated. Thus, I suggest adding ensemble validity check as well as reconsidering the computation of the model equivalent that binaiyly returns md or mw indistinctly of the water level value.**
See response to comment 21 for ensemble validity check. We agree that there are potential problems with applying the backscatter operator to a real case. We have added a new 'Discussion' section to address the problems noted here. See response to reviewer 1, comment 1.

25. **Locating the flood edge seems a difficult task for a real case with a randomly shaped flood surface, also, locating the nearest wet pixel is a complex task in non-idealized cases. I suggest to investigate this topic that is central to all 3 observation operator proposed here.**
We agree that locating the flood edge is a complex task; one of the advantages of the backscatter approach over the nearest wet pixel approach is that this step is not necessarily needed in order to perform data assimilation. We have made this clearer in the new 'Discussion' section – see response to reviewer 1, comment 1.

[revised manuscript text omitted]
 use the ETKF in its standard application as a sequential filter. As such we perform an update step at the time of each

15 available observation. We assume that the observations are related to the true state of the system, $\mathbf{x}^t$ according to

$$\mathbf{y}_{obs} = \mathbf{h}(\mathbf{x}^t) + \epsilon, \tag{5}$$

where the vector $\mathbf{y}_{obs} \in \mathbb{R}^p$ contains $p$ observations. The vector $\epsilon$ represents observation error, which we assumed to be unbiased and stochastic with covariance $\mathbf{R} \in \mathbb{R}^{p \times p}$. The observation operator, $\mathbf{h} : \mathbb{R}^N \to \mathbb{R}^p$ maps the state vector into observation space. If the state vector and the observation vector contain the same quantity (e.g. water depth) then the observation operator is

20 generally just required to pick out the values in the state vector corresponding to the spatial position of the observation(s); this may involve spatial interpolation if observations are not located at model grid points. However, it is commonly the case that observations are different quantities to those in the state vector and the observation operator therefore contains information about how the observed and state vector quantities are related as well as positional information. Different observation types (e.g. water elevation or wet/dry pixel information) therefore require different observation operators for the same model (i.e. for

25 the same state vector).

In order to update the model forecast it is useful to create a forecast-observation ensemble, which contains $M$ forecast-observation vectors, $\mathbf{y}_i^f$, $(i = 1, 2...M)$ such that

$$\mathbf{y}_i^f = \mathbf{h}(\mathbf{x}_i^f). \tag{6}$$

Equation (6) shows that the observation operator, $\mathbf{h}$, is applied to each state vector in order to extract observation equivalent information; each forecast-observation vector, $\mathbf{y}_i^f \in \mathbb{R}^p$ is what would be observed if the corresponding state vector, $\mathbf{x}_i^f$ represented the true state of the system. The model equivalent of the observation vector is given by the mean of the forecast-observation ensemble, $\overline{\mathbf{y}^f} \in \mathbb{R}^p$, where

$$\overline{\mathbf{y}^f} = \overline{\mathbf{h}(\mathbf{x})} = \frac{1}{M} \sum_{i=1}^{M} \mathbf{h}(\mathbf{x}_i). \tag{7}$$

Note that when the observation operator is nonlinear,

$$\overline{\mathbf{h}(\mathbf{x})} \neq \mathbf{h}(\overline{\mathbf{x}}). \tag{8}$$

We can also define a perturbation matrix $\mathbf{Y}^f \in \mathbb{R}^{p \times p}$ for the forecast-observation ensemble matrix:

$$\mathbf{Y} = \frac{1}{\sqrt{M-1}} (\mathbf{y}_1 - \overline{\mathbf{y}} \ \ \mathbf{y}_2 - \overline{\mathbf{x}} \ \ ...... \ \ \mathbf{y}_M - \overline{\mathbf{y}}). \tag{9}$$

The mean state vector and error perturbation matrix are updated separately in the ETKF. The mean state is updated according to

$$\overline{\mathbf{x}^a} = \overline{\mathbf{x}^f} + \mathbf{K}(\mathbf{y}_{obs} - \overline{\mathbf{y}^f}), \tag{10}$$

where $\overline{\mathbf{x}^a} \in \mathbb{R}^N$ and $\overline{\mathbf{x}^f} \in \mathbb{R}^N$ are the means of the analysis and forecast ensemble respectively. The ETKF uses an ensemble version of the Kalman gain, $\mathbf{K} \in \mathbb{R}^{N \times p}$ is, as defined in equation (13). The ensemble Kalman update (10) can be written in terms of the innovation, $\boldsymbol{\delta}_y$, where

$$\boldsymbol{\delta}_y = \mathbf{y}_{obs} - \overline{\mathbf{y}^f}. \tag{11}$$

The innovation is calculated in observation space. The term

$$\mathbf{K}(\boldsymbol{\delta}_y) \tag{12}$$

is known as the increment, and is the difference between $\overline{\mathbf{x}^a}$ and $\overline{\mathbf{x}^f}$. The increment is calculated in state space.

We use a square root formulation for the ETKF, following Ott et al. (2004), Livings et al. (2008) and Livings (2005). This formulation is also used in Garcia-Pintado et al. (2013) and Cooper et al. (2018). In this approach the ensemble version of $\mathbf{K}$ is written as

$$\mathbf{K} = \mathbf{X}^f (\mathbf{Y}^f)^T (\mathbf{Y}^f (\mathbf{Y}^f)^T + \mathbf{R})^{-1}. \tag{13}$$

The state error perturbation matrix is updated in the ETKF according to

$$\mathbf{X}^a = \mathbf{X}^f \mathbf{T}. \tag{14}$$

The perturbation matrix is updated by the matrix $\mathbf{T} \in \mathbb{R}^{M \times M}$.  We use an unbiased, symmetric square root formulation of the matrix $\mathbf{T}$, constructed in a way that ensures that the analysis state error covariance, $\mathbf{P}^a = \mathbf{X}^a(\mathbf{X}^a)^T$ is the same as the  analysis error covariance calculated in the Kalman covariance update (in e.g. Kalman (1960)).  The formulation makes use of a singular value decomposition (Golub and Van Loan (1996)),

$$(\mathbf{R}^{\frac{1}{2}}\mathbf{Y}^f)^T = \mathbf{U}\Sigma\mathbf{V}^T, \tag{15}$$

where $\mathbf{U} \in \mathbb{R}^{M \times M}$ and $\mathbf{V} \in \mathbb{R}^{p \times p}$ are orthogonal. The columns of $\mathbf{U}$ and $\mathbf{V}$ are the left and right singular vectors of $(\mathbf{R}^{\frac{1}{2}}\mathbf{Y}^f)^T$ respectively. The diagonal elements of the matrix $\Sigma \in \mathbb{R}^{M \times p}$ are the singular values of $(\mathbf{R}^{\frac{1}{2}}\mathbf{Y}^f)^T$. A solution for $\mathbf{T}$ is then

$$\mathbf{T} = \mathbf{U}(\mathbf{I} + \Sigma\Sigma^T)\mathbf{U}, \tag{16}$$

where $\mathbf{I}$ is the identity matrix. See Livings et al. (2008), Cooper et al. (2018) for  further details of how $\mathbf{T}$ is computed.

[revised manuscript text omitted]

A different approach to using binary-type observations in data assimilation is used by the authors of Rochoux (2014), Rochoux et al. (2014) and Rochoux et al. (2017) in an application in which the spread of wildfires is modelled. This approach uses shape recognition and front mapping; these ideas would be applicable to flood modelling but are not investigated here.

**4 Experimental design**

**4.1 Hydrodynamic model**

The inundation model used in this work is a non-linear hydrodynamic model. The model uses Clawpack code (Clawpack Development Team (2014), Mandli et al. (2016), LeVeque (2002)) to solve the two dimensional shallow water equations  everywhere in the domain, in order to simulate water flowing in a channel and overtopping onto a flood plain. Clawpack solves the shallow water equations using Riemann solvers and finite volume methods, and is able to simulate the wet-dry interfaces that occur during a flood George (2008). The software considers the domain of interest as a user-defined number of cells, $N$, and calculates changes in depth and velocity of the water in each cell. In our simulations the boundary condition is extrapolating (outflow) on the $y = 0$ boundary; all other boundaries are solid wall. Clawpack uses a source term in the momentum equation to model friction effects. Momentum reduction depends on a user-specified Manning's friction coefficient. Our experiments required an inflow source term to model water arriving in the river from upstream; we added this functionality to the Clawpack code, see Cooper et al. (2018) for details. The time step for the calculations is automatically adjusted to preserve numerical stability.

**4.2 Domain**

Experiments to compare the performance of the three operators have been carried out in an idealised river valley-like domain. The use of an idealised domain is important here so that we can examine the effects of the operators under ideal conditions, without the complications of complex topography. It will also be important to understand how the operators work under real conditions, but experiments in an idealised topography are a vital first step.

The test domain used in the experiments in this paper is the same as that used in Cooper et al. (2018) and is shown in figure 1. The domain has dimensions of 20km by 250m and describes a gently sloping valley and river channel (with upstream-downstream slope of 0.08%). The domain is split into grid cells of size 10m by 10m for computation. The river channel is prescribed to be the central 5 grid cells in the $x$ direction for all values of $y$ and is 50m wide; the flood plain is defined as the rest of the domain. The slope of the floodplain towards the river is 0.8% based on values derived from a DTM of a stretch of the river Severn in the U.K.

**4.3 Twin experiments**

We have carried out a number of twin experiments in order to illustrate and compare how well forecasts can be corrected when using the three different observation operator approaches. The experiments use a 'truth' flood simulation and a forecast

[Figure]

**Figure 1.** Test domain used in all assimilation experiments.

ensemble of flood realisations comprising 100 members. The forecast ensemble is updated using synthetic observations at several times during the simulation time; synthetic observations are created from the truth as described in section 4.4. The analysis water levels (and parameter values) can then be compared to the true water levels (and parameter values) to see how well the assimilation corrects the forecast.

5     In this work, the truth flood is driven by a time-varying inflow based on data taken from a gauge on the River Severn during a flood in November-December 2012. The true inflow is shown in figure 2; the figure also shows the inflows driving the ensemble members. All the inflows used here were also used in the experiments reported in Cooper et al. (2018). Inflows for each ensemble member were generated by perturbing the true inflow with additive, time correlated random errors. Time correlated errors were generated for each ensemble inflow using a first order autoregression (AR(1)) technique (Wilks (2011))

10   with zero mean, according to

$$e_{i,0} = w_{i,0},$$
$$e_{i,k} = re_{i,k-1} + (1 - r^2)w_{i,k}, \tag{23}$$

where $e_{i,k}$ is the error added to the inflow at the $k$th timestep in the $i$th ensemble member. The term $w_{i,k}$ is taken from a normal distribution $\mathcal{N}(0, 0.15 \times$ true inflow$)$; $i$ refers to ensemble member and $k$ refers to the timestep. The autocorrelation coefficient,

15   $r < 1$ was set to 0.997; this very high coefficient means that the errors are close to persistent in time for each ensemble member and that each inflow ensemble member is smooth. The standard deviation of the random part of the error corresponds to the value used to generate inflow errors in Garcia-Pintado et al. (2015) and results in inflows that fit within the range given in Di Baldassarre and Montanari (2009) (4% to 43%). The mean of the inflow ensemble has negligible bias relative to the true inflow. The experiments shown here all use the same inflow for the truth and the same set of perturbed inflows for the forecast

ensemble. For a different true inflow and different ensemble inflow error realisations, the results obtained using the different observation operators may compare slightly differently. However, the mechanisms we describe would be the same.

[Figure]

**Figure 2.** Inflows with time. True inflow values are represented with circles and ensemble inflows are shown by grey lines.

Each ensemble member was run with a different value of the channel friction parameter, $n_{ch}$. The behaviour of flood water is highly sensitive to $n_{ch}$ (Hostache et al. (2010), James et al. (2016)), with low channel friction parameter values leading to water travelling through and leaving the domain more quickly. This leads to shallower water levels (and less flooding) in our simple domain for a given inflow. Conversely, higher channel friction parameter values lead to water moving slowly through the domain, leading to deeper water levels in the channel and more flooding. We chose a true value of $n_{ch} = 0.04$, equal to the value for a natural stream given in Maidment and Mays (1988).  For the initial forecast step, a value of $n_{ch}$ for each forecast ensemble member was  drawn from a normal distribution with mean, $\mu$, that is different to the true value and standard deviation $\sigma$. This imposed bias in the forecast ensemble channel friction parameter means that we can test how well data assimilation with different observation operators can correct the forecast state and parameter value towards the truth.  In our state estimation experiments, the value of $n_{ch}$ assigned to each ensemble member remained constant throughout the simulation. For joint state-parameter experiments, the values of $n_{ch}$ were updated at each assimilation time through the ETKF equations, as described in section 2.2. Using an incorrectly specified 
[revised manuscript text omitted]

 In this study we wish to investigate the differences in the updates generated by different observation operator approaches. We therefore use equivalent observation information for each of the operators. In the case of the water level observation operators we have used flood edge water level  (observations at six locations,

[Figure]

**Figure 4.** Histograms and fitted Gaussian distributions of synthetic backscatter values. Dashed grey lines show two fitted Gaussian distributions and the solid grey line shows the sum of the two fitted distributions. Vertical lines show the positions of the mean wet and dry backscatter values.

where the flood edge location is defined as the  position of the first dry model cell (see section 4.4). For the  new operator we use two backscatter observations for each  5  transect.

[Figure]

**Figure 5.** Schematic of observation locations used in this study for each transect in cross section. The thick black line shows the discretised domain elevation, the dashed blue line shows the observed flood water level. The arrows and green crosses show locations of the observations as labelled.

Figure 5 shows a schematic of the locations of the observations we have used in this study, relative to the edge of the flood. All observations used in this study come from transects at $y = 500m, 700m, 900m, 1100m, 1300m$ and $1500m$. In practical application of the backscatter operator, observations could be used from any location covered by the SAR image.

**4.5 Observation error covariance matrices**

[revised manuscript text omitted]

**6 Discussion**

In this study we have chosen to use a small number of backscatter observations for our experiments. This allowed us to compare updates between the three observation operators when the observation operators were all given equivalent information; in this way we can draw conclusions about the physical mechanisms responsible for the different updates. In a real case, one of the major advantages of using our new backscatter observation operator is that it would be possible to use a large number of backscatter observations compared to the number of water level observations which are typically available. The availability of a large number of observations may be a major strength of our new approach; in our simple experiments (not shown) we found that assimilating a larger number of observations with the backscatter operator provided a better analysis than using only a few. Another merit of the backscatter operator is that there is less processing involved in using backscatter observations directly, potentially reducing the amount of time between acquisition of a SAR image and its use to update an inundation forecast. The backscatter operator also removes the need for locating the 'nearest wet pixel' in the model forecast, which can be computationally costly.

There are a number of potential problems with practical implementation of the backscatter operator. One is that using histograms to produce SAR-derived inundation maps can lead to errors in assigning pixels to wet/dry categories. One way to deal with this would be to use region growing techniques (see e.g. Horritt et al. (2001)) or change detection techniques (see e.g. Hostache et al. (2012)) to produce robust wet/dry maps for SAR images, and then perform a quality control procedure to discard any backscatter observations which would lead to mis-classification due to e.g. emergent vegetation. This procedure would remove the advantage of fewer processing steps for the backscatter operator, but may not be necessary. Further research is required to understand how robust the method is to the proportion of misclassified SAR pixels in a real case study. We note that the backscatter operator would not generate an update the forecast in model cells that all the ensemble members predicted to be dry (or wet) as discussed in the last paragraph of section 5.1.2. This means that SAR pixels far from the river wrongly classified as wet, or SAR pixels in the river channel wrongly classified as dry would not degrade the forecast through an erroneous update.

The new backscatter operator is likely to work well in cases where good separation of the wet/dry distributions can be obtained through a histogram, and less well in cases where the distributions overlap. The new observation operator does not require a digital elevation model to generate forecast-observation equivalents, although the hydrodynamic model would require

topography information to generate a forecast. Water level observations cannot be accurately determined in areas with high slope, whereas backscatter observations will be unaffected. Like the other observation operators, the new operator will likely provide better results in rural settings than urban settings; double-bounce and layover effects due to buildings are potential sources of problems for all of the operators (Mason et al. (2018)).

[revised manuscript text omitted]

Rochoux, Mélanie, C.: Towards a more comprehensive monitoring of wildfire spread : contributions of model evaluation and data assimilation strategies, Theses, Ecole Centrale Paris, https://tel.archives-ouvertes.fr/tel-01130329, 2014.

Rochoux, M., Collin, A., Zhang, C., Trouvé, A., Lucor, D., and Moireau, P.: Front shape similarity measure for shape-oriented sensitivity analysis and data assimilation for Eikonal equation, ESAIM: Proceedings and Surveys, pp. 1–22, https://hal.inria.fr/hal-01625575, 2017.

Rochoux, M. C., Ricci, S., Lucor, D., Cuenot, B., and Trouvé, A.: Towards predictive data-driven simulations of wildfire spread – Part I: Reduced-cost Ensemble Kalman Filter based on a Polynomial Chaos surrogate model for parameter estimation, Natural Hazards and Earth System Sciences, 14, 2951–2973, https://doi.org/10.5194/nhess-14-2951-2014, 2014.

Schumann, G., Bates, P. D., Horritt, M. S., Matgen, P., and Pappenberger, F.: Progress in integration of remote sensing derived flood extent and stage data and hydraulic models, Reviews of Geophysics, 47, https://doi.org/10.1029/2008RG000274, rG4001, 2009.

Smith, P. J., Dance, S. L., Baines, M. J., Nichols, N. K., and Scott, T. R.: Variational data assimilation for parameter estimation: application to a simple morphodynamic model, Ocean Dynamics, 59, 697, https://doi.org/10.1007/s10236-009-0205-6, 2009.

Smith, P. J., Dance, S. L., and Nichols, N. K.: A hybrid data assimilation scheme for model parameter estimation: application to morpho-dynamic modelling, Computers & Fluids, 46, 436–441, 10th ICFD Conference Series on Numerical Methods for Fluid Dynamics (ICFD 2010), 2011.

Smith, P. J., Thornhill, G. D., Dance, S. L., Lawless, A. S., Mason, D. C., and Nichols, N. K.: Data assimilation for state and parameter estimation: application to morphodynamic modelling, Quarterly Journal of the Royal Meteorological Society, 139, 314–327, 2013.

Stephens, E., Schumann, G., and Bates, P.: Problems with binary pattern measures for flood model evaluation, Hydrological Processes, 28, 4928–4937, https://doi.org/10.1002/hyp.9979, 2013.

Vörösmarty, C., Askew, A., Grabs, W., Barry, R., Birkett, C., Döll, P., Goodison, B., Hall, A., Jenne, R., Kitaev, L., Landwehr, J., Keeler, M., Leavesley, G., Schaake, J., Strzepek, K., Sundarvel, S., Takeuchi, K., and Webster, F.: Global water data: A newly endangered species, Eos, 82, 54+56+58, https://doi.org/10.1029/01EO00031, 2001.

Waller, J. A., García-Pintado, J., Mason, D. C., Dance, S. L., and Nichols, N. K.: Technical note: Analysis of observation uncertainty for flood assimilation and forecasting, Hydrology and Earth System Sciences Discussions, 2018, 1–13, https://doi.org/10.5194/hess-2018-43, 2018.

Wilks, D. S.: Statistical Methods in the Atmospheric Sciences, Academic Press, 2011.

Wood, M.: Improving hydraulic model parameterization using SAR data., Ph.D. thesis, University of Bristol, 2016.

Wood, M., Hostache, R., Neal, J., Wagener, T., Giustarini, L., Chini, M., Corato, G., Matgen, P., and Bates, P.: Calibration of channel depth and friction parameters in the LISFLOOD-FP hydraulic model using medium-resolution SAR data and identifiability techniques, Hydrology and Earth System Sciences, 20, 4983–4997, https://doi.org/10.5194/hess-20-4983-2016, 2016.